# Epigenetic modifications affect the rate of spontaneous mutations in a pathogenic fungus

Michael Habig [1,2✉], Cecile Lorrain[1,2], Alice Feurtey[1,2], Jovan Komluski[1,2] & Eva H. Stukenbrock [1,2✉]

Mutations are the source of genetic variation and the substrate for evolution. Genome-wide mutation rates appear to be affected by selection and are probably adaptive. Mutation rates are also known to vary along genomes, possibly in response to epigenetic modifications, but causality is only assumed. In this study we determine the direct impact of epigenetic modifications and temperature stress on mitotic mutation rates in a fungal pathogen using a mutation accumulation approach. Deletion mutants lacking epigenetic modifications confirm that histone mark H3K27me3 increases whereas H3K9me3 decreases the mutation rate. Furthermore, cytosine methylation in transposable elements (TE) increases the mutation rate 15-fold resulting in significantly less TE mobilization. Also accessory chromosomes have significantly higher mutation rates. Finally, we find that temperature stress substantially elevates the mutation rate. Taken together, we find that epigenetic modifications and environmental conditions modify the rate and the location of spontaneous mutations in the genome and alter its evolutionary trajectory.

[1] Environmental Genomics, Christian-Albrechts University of Kiel, Kiel, Germany. [2] Max Planck Institute for Evolutionary Biology, Plön, Germany. ✉email: mhabig@bot.uni-kiel.de; estukenbrock@bot.uni-kiel.de

Mutations generate heritable genetic variants upon which selection or drift can act. Therefore, detailed insights into the molecular processes that determine rates of mutation are central to our understanding of evolution[1]. In eukaryotes, DNA is organized in chromatin, which regulates transcription as well as DNA replication and DNA repair, and thereby appears to affect the mutation rate[2]. Posttranslational histone modifications shape the chromatin condensation state from euchromatin to heterochromatin[3]. In filamentous fungi trimethylation of lysine 9 of histone H3 (H3K9me3) is associated with constitutive heterochromatin and the suppression of transcription and transposition of transposable elements (TEs)[4]. In contrast, the trimethylation of the lysine 27 of histone H3 (H3K27me3) is associated with facultative heterochromatin that is enriched in the subtelomeric regions and in accessory genome compartments, and, in some species, also regulates gene expression[5–7]. Next to the role in transcriptional regulation, histone modifications also play a role in maintaining the integrity of the genome and its faithful transmission to daughter cells[8–11]. Moreover, chromatin organization was shown to correlate with the mutation rate in comparative genomic studies[10,12–14]. However, a causal resolution of this correlation by experimental manipulation has so far been missing. Therefore, it is also unknown if the observed correlation between mutation rate and chromatin organization or histone modification are causally linked directly—or if they both are influenced by a third factor (e.g., the local sequence context). In this aspect, it is crucial to understand whether the effect of histone modifications on mutation rates is due to their association with TEs or via DNA replication or DNA accessibility for mutagenic processes and/or DNA repair. Similarly, stress was shown to affect the mutation rate possibly by affecting similar mechanisms (i.e., activation of TEs, DNA replication, or endogenic or exogenic mutagenic processes)[15–18].

In general, most mutations in functionally relevant loci confer a deleterious or slightly deleterious effect, while only few mutations have a beneficial effect[19], and therefore the mutation rate itself is expected to be subject to selection[1]. The effectiveness of the selection processes on the mutation rate will not only depend on the effective population size of the organism[1] but may also be affected by the environment. Higher mutation rates may be selected for in changing or stressful environments[20]. Indeed, for many bacterial pathogens, hypermutator strains are frequently identified in clinical samples[21] as well as in microbial laboratory populations adapting to changing experimental conditions[20,22]. Host–pathogen co-evolution with reciprocal adaptations between the antagonists is highly dynamic and can be considered as frequently changing environments experienced by the pathogen[23]. Pathogens can adapt rapidly to new hosts, e.g., fungal plant pathogens can overcome plant resistance mechanisms or become resistant to fungicide within years[24,25]. High levels of genetic variation at the population level, in particular in genes that encode for virulence factors, appear to enable the adaption to changing host environments. In some species, rapid evolution is further accelerated in specific genomic compartments that are enriched with TEs or repetitive sequences, and are gene poor, but often encode virulence factors[26,27]. Evolution in these regions is often impacted by a variety of genome defense traits such as heterochromatin, repeat-induced point mutation (RIP) or DNA methylation, which are suggested to increase the mutation rate[28,29]. The clustering of virulence factors in genomic compartments more prone to mutations may confer higher rates of evolution in these genes, while facilitating the conservation of house-keeping genes in genomic compartments that are less prone to mutations[30]. In addition, structural variants are widespread in pathogenic fungi and are frequently associated with

pathogenicity-related genes in rapidly evolving genome compartments[31–33] as exemplified in the plant pathogenic species *Fusarium oxysporum* f.sp. *lycopersici*, *Nectria haematococca*, and members of the genus *Alternaria*. In these species, the main determinants of host specificity are located on accessory genomic elements[31–33]. Although variation of mutation rates between different genomic compartments of the genome are recognized as a possible mechanism that could facilitate rapid evolution of pathogenicity traits, experimental evidence is missing and the underlying determinants of the mutation rate are unknown.

The fungal wheat pathogen *Zymoseptoria tritici* is an ideal model to dissect mutation processes in different genomic compartments and the impact of epigenetic modifications and environmental stress on the mutation rate in a fungal plant pathogen. *Z. tritici* is a hemibiotrophic pathogen that infects wheat worldwide and causes dramatic losses in yield[34]. The bipartite genome of this fungus includes a set of 13 core chromosomes and a variable number of accessory chromosomes (i.e., eight in the reference isolate IPO323), which are subject to chromosome loss during mitosis and a meiotic drive during sexual reproduction[35–38]. In contrast to the yeast species *Saccharomyces cerevisiae* and *Schizosaccharomyces pombe*, *Z. tritici* possesses both H3K9me3 and H3K27me3 histone modifications —similar to the majority of plant and animal species. H3K9me3 mainly colocalizes with TEs, whereas H3K27me3 is enriched on the accessory chromosomes of *Z. tritici* (see Fig. 1a)[39]. Deletion strains for the responsible histone methyltransferases KMT1 and KMT6 are available and functionally characterized[39]. Field isolates of *Z. tritici* also differ in the presence of a functional DIM2, the methyltransferase responsible for cytosine methylation (5-methylcytosine (5mC)), and we could recently show that a functional DIM2 results in a higher rate of C → T transitions restricted to TEs[40]. A higher rate of transitions in TEs could be a defense mechanism against TE mobilization, but functional validation is missing[40]. The genome of *Z. tritici* appears highly variable[41] with TEs representing 18.6% of the genome[42] and 5% of the genome comprised introgressed regions[43], a genomic phenomenon that also exists in many other eukaryotic organisms[44]. TEs in recently introgressed regions may become unrepressed, a process that has been described for only few fungal species[45]. Interestingly, populations of *Z. tritici* comprise an unusually high standing variation with 44% of all nonsynonymous mutations estimated to be adaptive[46], which appears at odds with the basic tenet that most mutations in functionally relevant loci are assumed to confer a deleterious or slightly deleterious effect.

In this study, we therefore aimed to (i) quantitatively assess the impact of H3K9me3 and H3K27me3 on the mutation rate in *Z. tritici*; (ii) determine the effect of 5mC DNA methylation on the mobilization of TEs; (iii) understand the changes in the mutation rate at mild temperature stress, at a level that is likely to be encountered during the life cycle of the organism; (iv) investigate the mobilization of TEs, in particular in introgressed regions of the genome; and (v) understand whether the fitness effects of spontaneous mutations could explain the high rate of adaptive mutations observed in populations of *Z. tritici*. We addressed these aims employing mutation accumulation (MA) experiments that use strong and repeated bottlenecks in highly replicated settings to maximize drift and minimize selection, which allowed us to determine and compare mutation rates along the fungal genome and among strains. We show that in *Z. tritici* H3K27me3 increases, whereas H3K9me3 decreases the mutation rate, while the accessory chromosomes overall have a significantly higher mutation rate than the core chromosomes. In addition, 5mC DNA methylation increases the mutation rate substantially and results in less TE mobilization. The mutation rate also increases

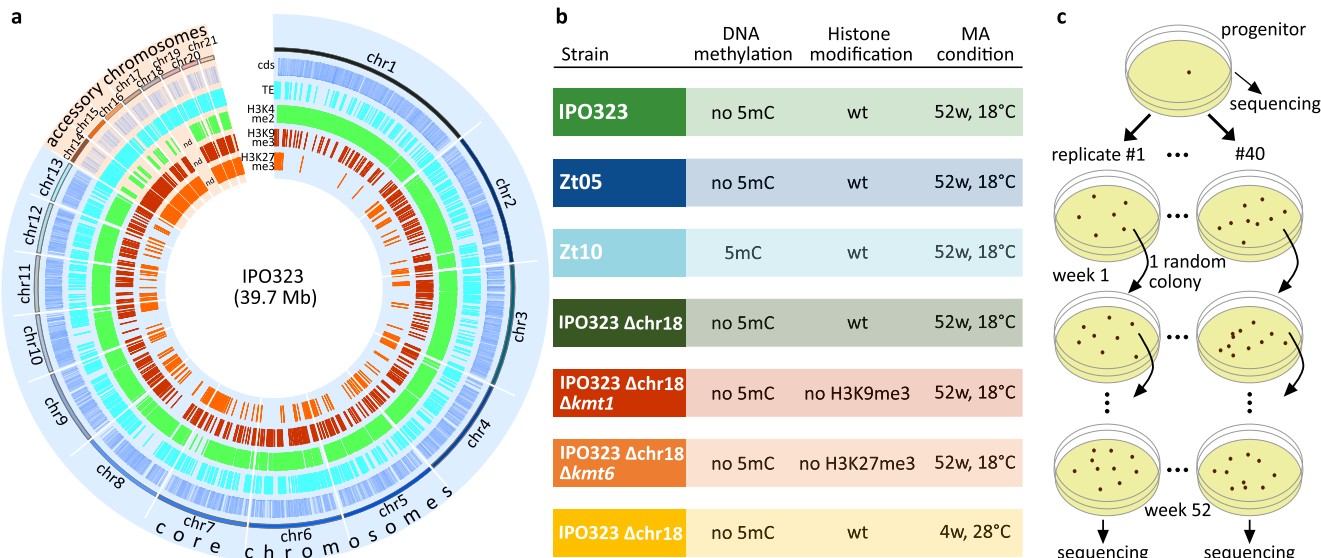

**Fig. 1 Overview of the genomic features and the experimental procedures to determine the effect of DNA methylation, histone modifications, and temperature stress on mutation rates. a** Circosplot of the genomic, genetic, and epigenetic features of the reference isolate IPO323 and their distribution among the 13 core and 8 accessory chromosomes (blue: coding sequences (cds), aquamarine: transposable elements (TEs), green: H3K4me2, red: H3K9me3, and orange: H3K27me3 (data from ref. [39]). **b** Experimental conditions for the seven genotype × environment combinations, each comprising 40 independently evolved replicates (MA: mutation accumulation, color code used throughout the study; wt: wild type; w: week). **c** Experimental procedure for one genotype × environment as an example. Each week, a random one-cell bottleneck was introduced. See Supplementary Methods for a detailed description of the experimental procedures.

upon mild temperature stress. MA and similar evolution experiments have mostly been restricted to model organisms in standard environments[18,47–55]. This study is one of the few that directly determine the impact of epigenetic modifications and temperature stress on the mutation rate in a pathogen with a highly structured, bipartite genome.

## Results

**The mutation rate is higher in accessory compartments and correlates with histone modifications.** To assess the mutation rate variation along the genome of *Z. tritici*, we conducted an MA experiment of seven genotype × environment combinations for 52 or 4 weeks, each with 40 independent replicated MA lines (Fig. 1).

We compared the genomes of progenitor and evolved strains by whole genome sequencing and in-depth analyses of mutational events (see Supplementary Methods for detailed description of the MA experiment). During propagation, we introduced a single-cell bottleneck each week, to maximize drift and minimize the confounding impact of selection. We identified a total of 14,834 sequence alterations in the genomes of the 280 evolved replicated MA lines compared to their respective progenitor (Supplementary Table 1). A total of 9813 spontaneous mutations were single base substitutions, whereas 5021 mutations affected more than 1 bp. Different genotypes and treatments with their differences in growth rates and number of mutable sites were compared by normalizing the number of observed mutations with the total number of cell divisions and the number of mutable sites (see Supplementary Methods). We computed the overall base substitution rate ($\mu$) to be $3.2 \times 10^{-10}$ and $3.01 \times 10^{-10}$ (per site per cell division) during propagation for 52 weeks at 18 °C in the reference strain IPO323 and IPO323 Δchr18, respectively. This base substitution mutation rate increased ~5-fold to $\mu = 16.2 \times 10^{-10}$ when the fungus was propagated at 28 °C (Fig. 2). Next to IPO323, we included two additional field isolates, Zt05 and Zt10, collected in Denmark and Iran, respectively, in the MA experiment. Zt05 had a similar base substitution mutation rate of $\mu = 2.90 \times 10^{-10}$, whereas the isolate Zt10 had a markedly

different rate of $\mu = 45.17 \times 10^{-10}$. This ~15-fold increase in the mutation rate in Zt10 is caused by an increase in C → T transitions, which occurred mainly in TEs. These mutational spectra alone (i.e., in the absence of other factor(s) affecting the GC content) would result in an equilibrium GC content of 52.6% in IPO323 after 52 weeks at 18 °C and 51.8% in Zt05 (Fig. 2b). These are very similar to the observed GC content in IPO323 and Zt05 at 52.1% and 51.9%, respectively. Zt10, again, markedly differed. Here, the high rate of G:C > A:T transitions, which are restricted mainly to TEs, would result in an equilibrium genome-wide GC content of 2.7%, compared to an observed GC content of 51.8%.

Mutations were not randomly distributed along the genome (Supplementary Fig. 1). To identify effects of genomic location, we correlated the occurrence of mutations to different genetic and epigenetic features. To this end, we used the wild type strain IPO323 and the derived strain IPO323 Δchr18 for which annotations of genes and TEs, as well as the distributions of histone methylation H3K4me2, H3K9me3, and H3K27me3 are available (see Fig. 1)[39,42]. In our experiment, the evolved replicated MA lines showed a significantly higher mutation rate on the accessory chromosomes than on the core chromosomes, in both IPO323 and IPO323 Δchr18 (see Supplementary Data 1 for results of all statistical analyses) (Fig. 3a). In addition, we identified specific genomic compartments, which also differed in their mutation rate. We found that the rate of base substitution mutations was significantly reduced in coding sequences and regions colocalizing with H3K4me2 (Fig. 3b), which is associated with transcriptionally active euchromatin in fungi, including *Z. tritici*[11,56]. In contrast, significantly higher mutation rates were observed in TEs and regions colocalizing with either of the histone modifications H3K9me3 or H3K27me3. However, as these genomics features partially overlap, the above-mentioned effects could be due to a combinational effect of the individual factors. We therefore conducted a full-factorial analysis for the genomic features of TEs, H3K9me3, and H3K27, and determined the mutation rate for any possible combination of these features. The total size of some regions with a particular combination of genetic

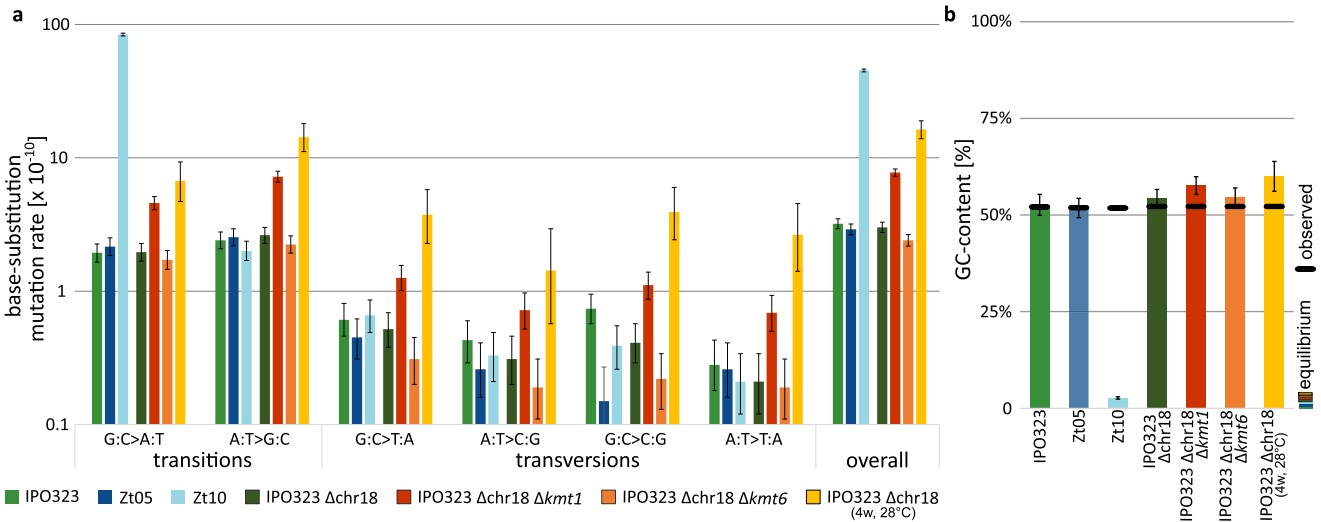

**Fig. 2 The mutation spectrum in *Z. tritici* is affected by epigenetic modification and temperature. a** Rates of base substitution mutations according to transitions and transversions. Comparison between *Z. tritici* wild type strains (IPO323, Zt05, and Zt10), which differ for DIM2-mediated 5mC methylations (Zt10 shows DIM2-mediated 5mC methylations; IPO323 and Zt05 do not) and comparison between *Z. tritici* strains (IPO323 Δchr18, IPO323 Δchr18 Δkmt1, and IPO323 Δchr18 Δkmt6), which differ in the presence of histone modifications H3K9me3 and H3K27me3, respectively. In addition, the effect of an increase in temperature from 18 °C to 28 °C on the rate of base substitution mutations is depicted. G:C > A:T summarizes G → A and C → T base substitution mutations (analogous in the other depicted categories). Data were pooled for all replicated MA lines ($n = 40$ independently evolved MA lines for each strain) and is presented as mean values. Error bars represent 95% Poisson confidence intervals. **b** Equilibrium GC content estimated on the rates of mutations in AT direction (G:C > A:T transitions and G:C > T:A transversions) and the rate of the mutations in GC direction (A:T > G:C transitions and A:T > C:G transversions), compared to the observed GC content in the respective reference genomes. Data were pooled for all replicated MA lines ($n = 40$ independently evolved MA lines for each strain) and is presented as mean values. Error bars represent the SE as described in Supplementary Methods.

and epigenetic features was particularly small (e.g., H3K27me3 ∩ TE − H3K9me3 = 51,269 bp) and observing mutations in these small regions in individual replicated MA lines was rare. Therefore, data were pooled for all replicate lines of both the IPO323 and the IPO323 Δchr18, to increase statistical power of the comparison of the mutation rate between genomic compartments.

H3K27me3 and TEs, but not H3K9me3, were associated with a significantly higher base substitution mutation rate. The lowest base substitution mutation rate ($1.96 × 10^{-10}$) was observed in regions that do not colocalize with H3K9me3, H3K27me3, or TEs. However, there was no significant difference in the base substitution mutation rate in regions of the genome that colocalize with H3K9me3 alone. In addition, the highest base substitution mutation rates ($11.8 × 10^{-10}$) was found in those regions that colocalize with all three features (H3K9me3, H3K27me3, and TE). Small INDELs (insertions and deletions, ≥ 2 bp) were also affected by the genomic features. Notably, we observed a particularly high rate of INDELs on chromosome 14 and a lower rate in coding sequences and regions colocalizing with H3K4me2 (Supplementary Fig. 2a, b). In contrast to base substitutions, we did not observe a consistent correlation between INDEL and H3K9me3, H3K27me3, or TEs.

Similarly, we found that the occurrence of larger structural variants (≥10 bp) was not random across the genome (Fig. 4a and Supplementary Figs. 3 and 4). The frequency of deletions, insertions, and duplications was increased in regions with TEs, H3K9me3, and/or H3K27me3 as compared to intergenic regions without TEs (Supplementary Fig. 3). In the full-factorial analysis, no single epigenetic or genetic feature was associated with a significant change in the frequency of deletions (Fig. 4b). Rates of deletions, insertions, and duplications were, in general, significantly increased in regions of the genome that colocalize with two or more of the epigenetic and genetic features, which could indicate a synergistic effect between these factors.

**Removal of H3K9me3 increases while removal of H3K27me3 decreases base substitution mutation rate**. To test the hypothesis that histone modifications modify the mutation rate, we took advantage of two *Z. tritici* mutants that are impaired in heterochromatin formation, Δkmt1 (no H3K9me3) and Δkmt6 (no H3K27me3), and included these in the MA study and propagated them, as described above, for 52 weeks. The removal of H3K9me3 significantly increased the genome-wide base substitution mutation rate. In contrast, the absence of H3K27me3 significantly decreased the genome-wide base substitution mutation rate (Fig. 5a), confirming that H3K27me3 promotes mutations. In the full-factorial analysis, we found that removal of H3K9me3 increased the mutation rate in each of the possible eight combinations of the three features (TEs, H3K9me3, and H3K27me3) compared to the wild type. The removal of H3K27me3, however, resulted in only one significantly reduced mutation rate compared to the wild type—in regions that were associated with TEs, H3K9me3, and H3K27me3. For all other combinations of the three features, no significant difference in the mutation rate compared to the wild type was observed (Fig. 5b).

Heterochromatin provides an important mechanism to silence TE and to maintain genome stability[11,57]. In *Z. tritici*, heterochromatin furthermore plays a role in the stability of accessory chromosomes[58]. Genome data from the evolved Δkmt6 and Δkmt1 mutants provides a unique opportunity to quantitatively assess the impact of H3K9me3 and H3K27me3 on TE mobilization and chromosome stability. To this end, we determined the extent of structural variants and chromosome stability in the evolved genomes of the mutants. Indeed, we find that histone marks have a significant effect on the rate of structural variants and the stability and replication of chromosomes. Specifically, removal of H3K9me3 affected the transmission of the entire accessory chromosomes. Among the 40 replicate lines lacking

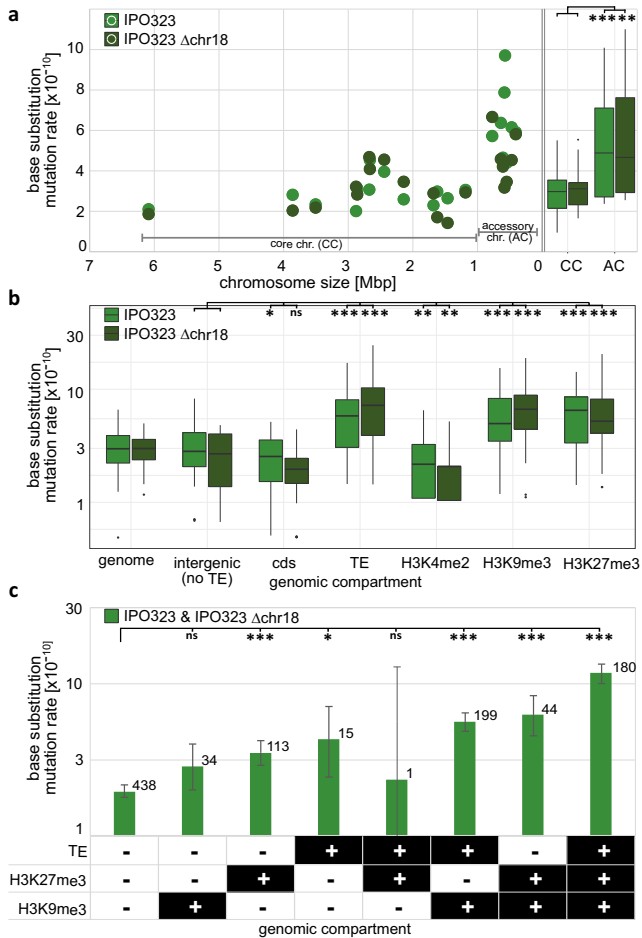

**Fig. 3 Accessory chromosomes, the presence of histone modifications, and TEs are associated with a higher base substitution mutation rate in the wild type. a** Correlation between the mean mutation rate for each of the chromosomes for both IPO323 (light green) and IPO323 Δchr18 (dark green) after 52 weeks at 18 °C, also including a boxplot comparing the base substitution mutation rates within all replicated MA lines between core chromosomes (CCs) and accessory chromosomes (ACs) ($n = 40$ independently evolved MA lines for each strain). **b** Boxplot comparing the base substitution mutation rates within each replicated MA line of IPO323 and IPO323 Δchr18 ($n = 40$ independently evolved MA lines for each strain). **c** Comparison of the base substitution mutation rates in a full-factorial approach for the three genomic features TE, H3K9me3, and H3K27me3. Data were pooled for all replicated MA lines of IPO323 and IPO323 Δchr18 ($n = 80$ independently evolved MA lines) and is presented as mean values. Total number of observed base substitutions in each genomic compartment is depicted above each bar. Error bars represent 95% Poisson confidence intervals. Categorized FDR-corrected $p$-values of **a**, **b** two-sided paired Wilcoxon tests or **c** two-sided $\chi^2$-tests are shown (*$p < 0.05$, **$p < 0.005$, ***$p < 0.0005$). In order to increase clarity here, only the results of pairwise comparisons that are discussed in the text are depicted. The exact $p$-values for all pairwise comparisons are provided in Supplementary Data 1. **a**, **b** Box plots depict center line, median; box limits, upper and lower quartiles; whiskers, 1.5× interquartile range; points, outliers.

H3K9me3, a total of 53 independent events of accessory chromosome losses occurred during the 52 weeks of mitotic cell divisions, significantly more than in the wild type strain (Fig. 5d). Interestingly, this increase was mainly due to extensive losses of the two specific accessory chromosomes chr14 and chr20 (Supplementary Fig. 5a). In addition, duplications of chromosomes affecting both core and accessory chromosomes occurred

at a higher frequency in the strains lacking H3K9me3 (Supplementary Fig. 5a, b). The removal of H3K9me3 also led to a significant increase of deletions and duplications also in regions not covered by H3K9me3 in the wild type (Supplementary Fig. 6a, b). In contrast, the removal of H3K9me3 led to a significant reduction of insertions and the few insertions that did occur were specific to regions otherwise not colocalizing with the H3K9me3 in the wild type strain (Fig. 5e and Supplementary Fig. 6b). Larger duplications (>10 kb) in Δkmt1 clustered on chr1, chr8, chr9, and chr12. In all cases, the affected regions are limited by a TE on one side and the centromeric region on the other side —a pattern similar to the one observed in a recent study[40]. Nevertheless, it is important to note that many smaller duplications also occur within these regions (see Supplementary Data 5).

For the 40 replicate lines lacking H3K27me3, we found, opposite to the evolved Δkmt1 lines that lack H3K9me3, a significantly reduced rate of accessory chromosome losses compared to the wild type. Interestingly, the lack of H3K27me3 had no significant effect on the number of duplicated chromosomes, the overall number of structural variants, or the rate of structural variants (Fig. 4d, e). In conclusion, H3K9me3 and H3K27me3 affected the mitotic transmission of chromosomes in opposite ways, but the increase in loss of accessory chromosomes observed when H3K9me3 was removed did not appear to be due to an increase in TE mobilization.

**DNA methylation affects the frequency of structural variants and TE mobilization.** Next, we assessed the effect of DNA methylation as another type of epigenetic modification on genome stability. We took advantage of naturally occurring variation in DNA methylation among field isolates of *Z. tritici*. Next to IPO323, we included two additional field isolates, Zt05 and Zt10, in the MA experiment. These strains differ in the presence of a functional DIM2 DNA methyltransferase and active cytosine DNA methylation (5mC)[40,59]. IPO323 and Zt05 lack 5mC, whereas the genome of Zt10 encodes a functional *dim2* gene and shows 5mC[40,59]. 5mC in Zt10 is associated with a high level of C → T transitions—specifically at CpA sites in TEs. In IPO323 and Zt05, no such high levels of transitions are present[40]. The increased level of C → T transitions in TEs caused by the presence of the *dim2* gene is likely a mechanism to inactivate TEs[40]. Here we expanded on the previous analysis to include structural variants in order to test whether the presence of 5mC is associated with a lower TE mobilization, which would be expected if 5mC in *Z. tritici* is a defense mechanism against TE mobilization. Indeed, the significantly increased rate of base substitution in Zt10 was associated with a lower rate of structural variation, such as deletions, insertions, and duplications (Fig. 6a). In addition, the association of TEs with deletions was reduced (Fig. 6b). Hence, 5mC appears to affect the frequency of structural variation and the involvement of TEs.

**Temperature stress increases the base substitution mutation rate and the rate of structural variants.** Next, we tested whether mild temperature stress - within the range that can be encountered by the pathogen in the field - had an effect on the base substitution rates and the rate of structural variation in *Z. tritici*. In line with our hypothesis, the base substitution mutation rate increased more than five-fold from $\mu = 3.01 \times 10^{-10}$ at 18 °C to $\mu = 1.6 \times 10^{-9}$ at 28 °C affecting all genomic compartments (Fig. 5a). Interestingly, the largest increase in the base substitution mutation rate was ~35-fold in regions that colocalize with both histone marks H3K9me3 and H3K27me3 (Fig. 5b), indicating an interaction between histone modification and stress in promoting

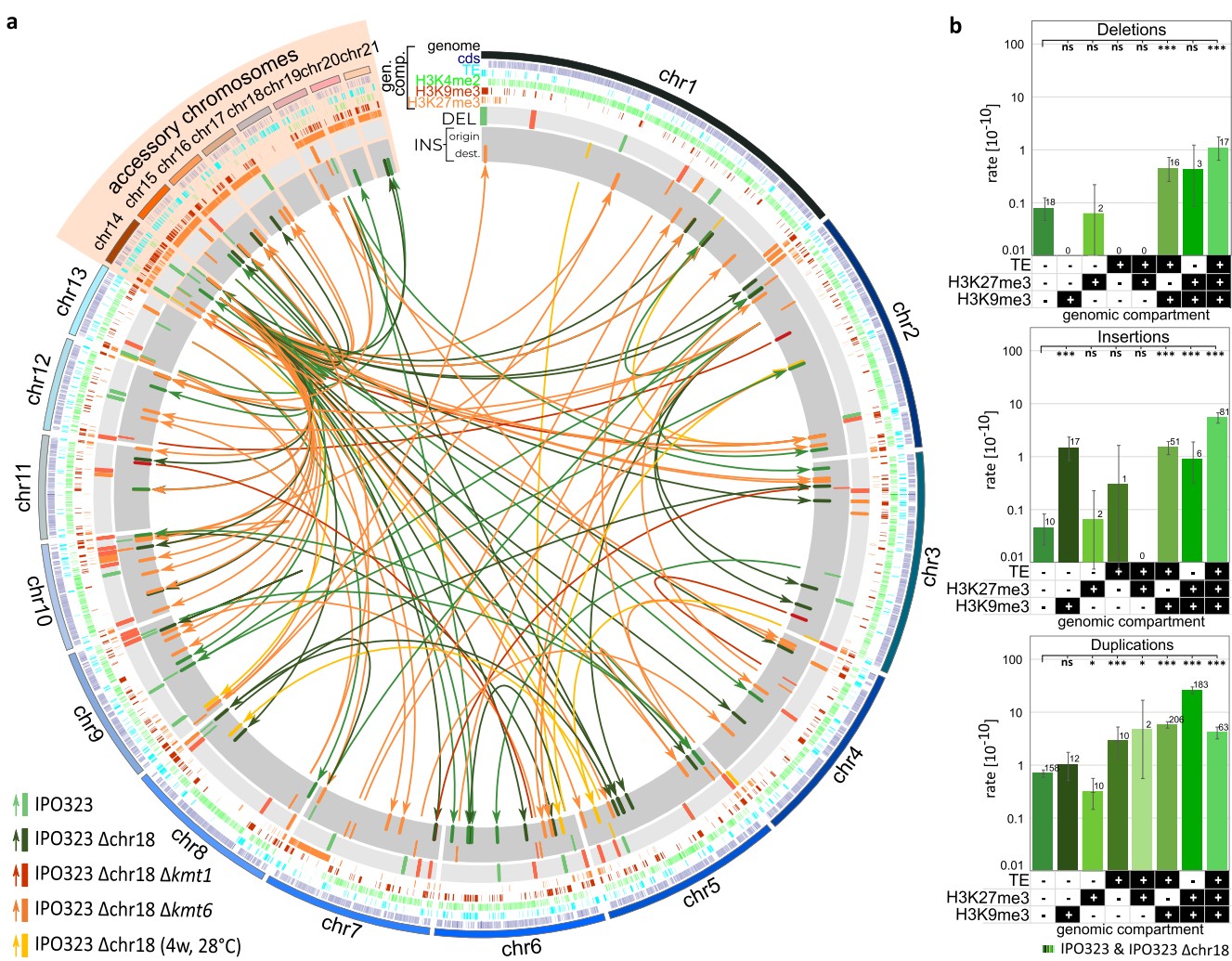

**Fig. 4 Histone modifications and TEs are associated with the occurrence of structural variants. a** Circosplot of the locations of insertions (on dark gray) and deletions (on light gray) in IPO323-derived strains. DEL = deletions, INS = insertions. For insertions, the origin and the destination of insertions were determined. Connections indicate the location of the corresponding origin and destination of insertions. In addition, the locations of cds, TE, and the posttranslational histone marks H3K4me2, H3K9me3, and H3K27me3 are given. **b** Detailed analysis of the occurrence of start and end positions of structural variants in different genomic compartments. Data were pooled for all replicated MA lines of IPO323 and IPO323 Δchr18 ($n = 80$ independently evolved MA lines) and is presented as mean values. Categorized $p$-values of two-sided $\chi^2$-test corrected by FDR for multiple testing are shown. (*$p < 0.05$, **$p < 0.005$, ***$p < 0.0005$). In order to increase clarity, only the results of pairwise comparisons that are discussed in the text are depicted. The exact $p$-values for of all pairwise comparisons are provided in Supplementary Data 1. Error bars represent 95% Poisson confidence intervals. Total number of observed structural variants is depicted above each bar.

mutations. Moreover, we find that temperature stress changed the mutational spectrum, lowering the Ts/Tv ratio from 3.1 at 18 °C to 1.7 at 28 °C, which could be indicative of differences in the mutational processes (Fig. 4c). The rate of some structural variants was also significantly increased at 28 °C (Fig. 5d, e)—the rate of accessory chromosome losses and chromosome duplications, and the rate of deletions and duplications was increased—but interestingly, not the rate of insertions (Fig. 5d, e).

**Recently integrated genomic regions are a source for activated TEs.** *Z. tritici* contains highly variable regions throughout the genome reflecting past interspecies hybridization events[43]. In other fungal species, such recently integrated genomic regions are considered to be a source of active TEs[45]. Using the evolved *Z. tritici* strains from our MA experiment, we asked whether the most active TEs are associated with more recently introduced genomic regions. To this end, we mapped inserted sequences to

their original genomic location. In IPO323 and the derived strains, IPO323 Δchr18, IPO323 Δchr18 Δ*kmt1*, and IPO323 Δchr18 Δ*kmt6*, we detected a total of 118 insertions (Fig. 4a). Interestingly, of these 118 insertions, 62 originated from chromosome 14, and of these, 57 within a region of chromosome 14, which shows the presence/absence of polymorphism between different field isolates of *Z. tritici* and is absent in sister species[60]. We speculate that this region represents a recent invasion in the *Z. tritici* genome possibly via introgression from another, unknown species. Within this region on chromosome 14, 53 insertions mapped to one location that comprises a retrotransposon belonging to the LINE superfamily (ID = ms2365). The inserts originated from the 3′-end of the retrotransposon and vary in length from 141 to 555 bp (Supplementary Fig. 7). This is consistent with the expected pattern for transposition of LINE retrotransposon for which partial transpositions are frequently observed due to aborted reverse transcription primed by DNA/RNA pairing at the poly(A)-tail of the transposon RNA[61]. The

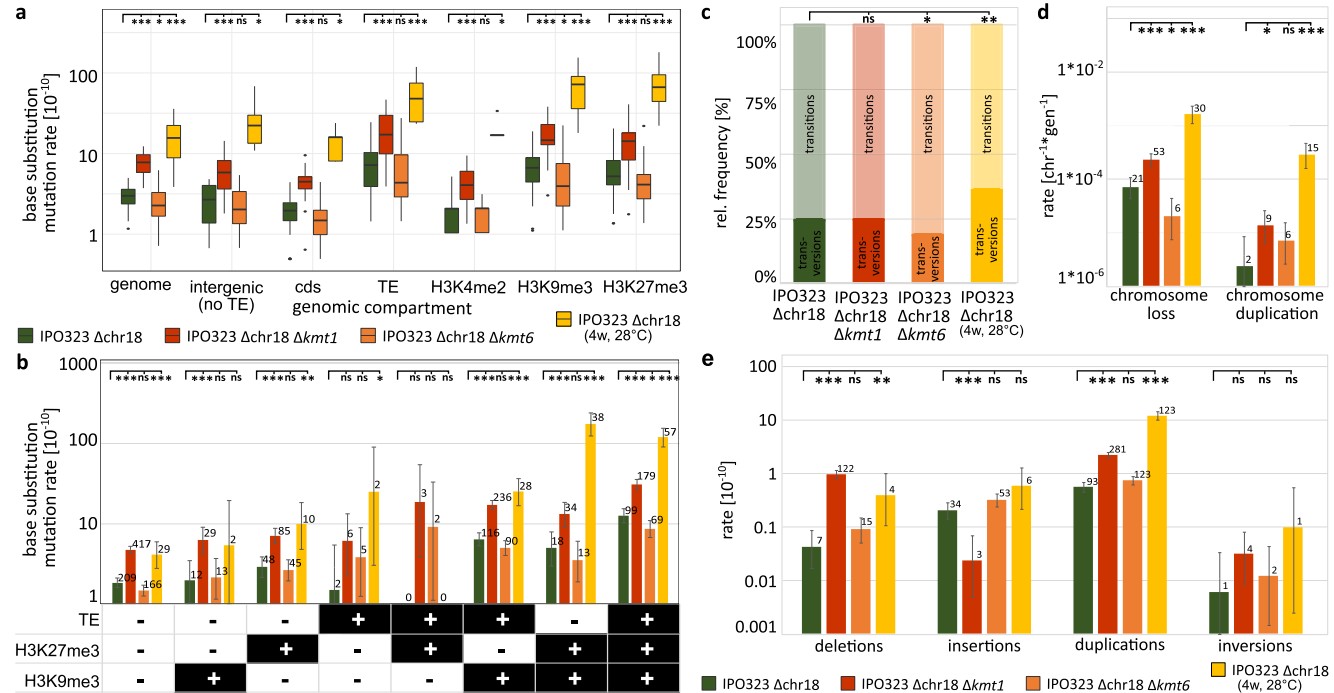

**Fig. 5 The removal of the histone modifications H3K9me3 and H3K27me3, and an increase in temperature affect the rate of mutations differently between different genomic compartments. a** Boxplot comparing the rate of base substitution rates in different compartments for all replicated MA lines of IPO323 Δchr18 and IPO323 Δchr18 Δkmt1 (lacking H3K9me3), and IPO323 Δchr18 Δkmt6 (lacking H3K27me3) propagated at 18 °C for 52 weeks and IPO323 Δchr18 propagated for 4 weeks at 28 °C (n = 40 independently evolved MA lines per strain). **b** Detailed analysis of the effect of the mutation rate in genomic compartments associated with TE, H3K9me3, and H3K27me3 using pooled data of all replicated MA lines for each strain. Effects of the removal of the respective histone modification and increase in temperature **c** on the relative frequency of transversions and transitions, **d** on whole chromosome aberrations, and **e** on structural variants. The genomic compartments are defined by the presence of the indicated genetic or epigenetic feature in the wild type IPO323. Categorized p-values of **a** FDR-adjusted two-sided Wilcoxon signed rank test or **b**–**e** two-sided Fisher's exact test or FDR-adjusted two-sided χ²-test are depicted (*p < 0.05, **p < 0.005, ***p < 0.0005, ns = not significant at α = 0.05). The exact p-values for of all pairwise comparisons are provided in Supplementary Data 1. **b**, **d**, **e** Data were pooled for all replicated MA lines (n = 40) of each strain and is presented as mean values. Error bars depict 95% Poisson confidence intervals. **a** Box plots depict center line, median; box limits, upper and lower quartiles; whiskers, 1.5× interquartile range; points, outliers.

observed pattern therefore indicates the presence of an active retrotransposon in a recently integrated genomic region of chromosome 14.

**Accumulated mutations vary in fitness effects but are on average deleterious.** A basic tenet of evolutionary biology is that most functionally relevant mutations are neutral or deleterious in any given environment[62]. In Z. tritici, a high proportion of nonsynonymous mutations appear adaptive, implying a strong effect of natural selection in removing non-advantageous mutations and fixing the beneficial ones[46]. The evolved Z. tritici strains allowed us to test whether the spontaneous mutations have a beneficial or a deleterious fitness effect. We therefore compared the fitness of evolved replicated MA lines with the respective progenitor using the ability to produce pycnidia (asexual fruiting bodies) as a proxy for fitness[37]. We set up an infection experiment using the three wild type isolates, IPO323 (IPO323 Δchr18), Zt05, and Zt10, which all cause disease on the same susceptible wheat cultivar. Based on the number of pycnidia produced, we find that the pooled data for all evolved MA replicate lines showed a lower median in planta fitness compared to the progenitor strains (with significantly different pycnidia densities, for three of the five tested strains, Fig. 7). Interestingly, the ability of the replicated MA lines to produce pycnidia varied considerably, with a number of them having a significantly reduced ability to produce pycnidia in planta compared to the progenitor strain (Supplementary Fig. 8a–e). The evolved replicated MA lines also

varied in their ability to grow in vitro as exemplified by the 40 replicated MA lines of the IPO323 Δchr18 (4w, 28 °C). These varied in tolerance to five cell stressors (Supplementary Fig. 9) and 15 of the 40 evolved replicated MA lines had a markedly reduced carrying capacity during growth in liquid medium (Supplementary Fig. 10a). The carrying capacity should represent their ability to utilize the nutrients in the media and therefore their metabolic potential (Supplementary Fig. 10a). Interestingly, we find an overall correlation between the carrying capacity in vitro and the in planta fitness supporting the negative effect of the accumulated mutations (Supplementary Fig. 10b). Hence, the majority of the accumulated mutations have a negative or neutral fitness effect in vitro and in planta, with only one replicate line (replicate 38 in IPO323) showing a significantly higher pycnidia density and thereby fitness in planta.

## Discussion

Here we establish the mutation rate in a pathogenic fungus, correlate the observed mutation rates with epigenetic histone and DNA modifications, and evaluate the effect of removing these epigenetic modifications on the mutation rates. Moreover, we assess the impact of mild temperature stress on mutation rates along the fungal genome. Our study shows, with experimental data so far unprecedented for any other species, that different epigenetic modifications directly affect the mutation rate.

The base substitution mutation rate of $\mu = 2.9$–$3.2 \times 10^{-10}$ per site per cell division observed in the wild type IPO323 and Zt05 at

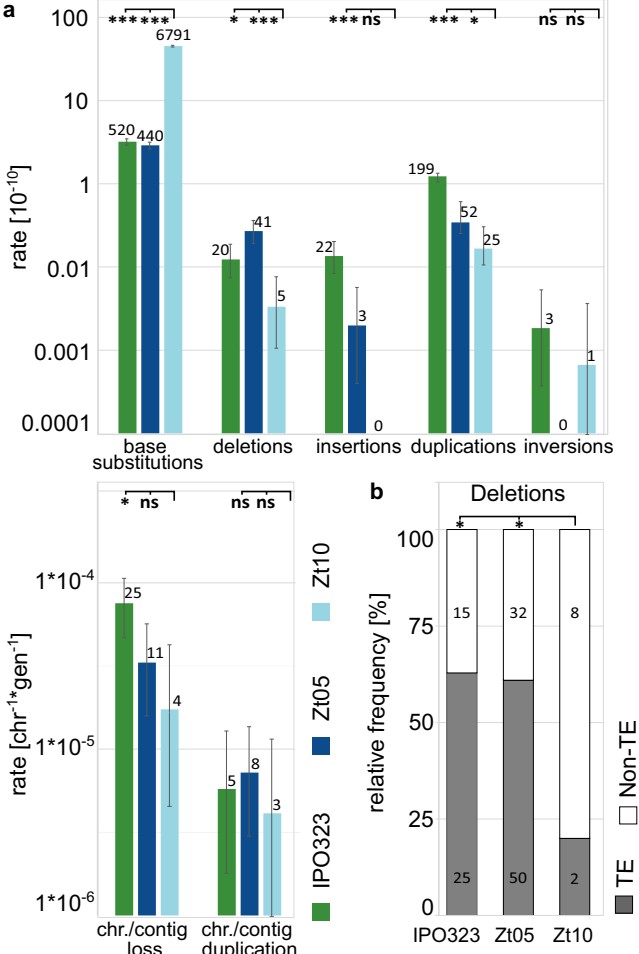

**Fig. 6 A functional *dim2* gene is associated with less structural variation and less effects of TE on deletions.** Differences in the rate of structural variants in wild type strains that do (Zt10) or do not (IPO323 and Zt05) contain a functional *dim2* gene. **a** Rate of base substitution mutation and structural variants, and chromosome losses and duplications in the evolved replicated MA lines. The significantly increased base substitution mutation rate in Zt10 (functional *dim2*) is associated with significantly lower rates of deletions, insertions, and duplications, and lower rates for chromosome losses and duplications. **b** Relative frequency for the start and end positions of deletions being associated with TEs is lower for deletions in Zt10 compared to deletions in IPO323 and Zt05, which lack a functional copy of the *dim2* gene. Categorized *p*-values of **a** FDR-adjusted two-sided $\chi^2$-test or **b** two-sided Fisher's exact test are depicted (*$p < 0.05$, **$p < 0.005$, ***$p < 0.0005$, ns = not significant at $\alpha = 0.05$). The exact *p*-values for of all pairwise comparisons are provided in Supplementary Data 1. **a** Data were pooled for all replicated MA lines of each strain (*n* = 40 independently evolved MA lines per strain) and is presented as mean values. Error bars depict 95% Poisson confidence intervals.

18 °C is similar to the previously reported $\mu = 1.67–4.04 \times 10^{-10}$ and $\mu = 1.7–2.0 \times 10^{-10}$ for the two model ascomycete yeast species *S. cerevisiae*[49,63–65] and *S. pombe*[52,53], respectively, as well as to base substitution rates assessed in other ascomycetes and basidiomycetes yeasts[47,55]. However, although the absolute mutation rate is similar to the rates reported in other ascomycetes, the mutation spectrum with high rate of A → G and T → C transversions differs markedly in *Z. tritici* compared to other ascomycetes, resulting in considerably higher Ts/Tv ratios, more similar to a mutation spectrum reported in the basidiomycete yeast *Rhodotorula toruloides*[47]. Likewise, in contrast to previously reported mutation spectra in eukaryotes, we do not see a A/T mutation bias in those isolates that lack 5mC[66], suggesting that other mechanisms are at stake for the occurrence of these mutations.

Interestingly, we find that the GC content in field isolates of *Z. tritici* that lack DNA methylation is at equilibrium, unlike in any other previously described fungus, implying that the mutation spectrum alone may explain the observed GC content. In many other species, deviations from the mutational GC equilibrium are caused by GC-biased gene conversion[67,68]. Interestingly, GC-biased gene conversion was not detected in *Z. tritici*, based on analyses of population genomic data[69], which further support the lack of deviations from the mutational equilibrium GC content observed here. The majority of the accumulated functional mutations had a neutral or deleterious effect on fitness during in vitro and in planta propagation in our experiments, similar to the observed fitness effects in other organisms (reviewed in ref. [19]).

Accumulated mutations were not randomly distributed along the genome. A unique feature of *Z. tritici* is the large complement of accessory chromosomes, which may represent selfish elements that employ the meiotic machinery for their propagation[36]. Here we show that accessory chromosomes have a higher base substitution mutation rate compared to core chromosomes during mitotic growth. In particular, this higher mutation rate is independent of the proposed effect of the fungal defense mechanism RIP, which occurs during meiotic but not mitotic propagation[30]. The location of genes under diversifying selection—e.g., genes manipulating the host immune system—could profit from

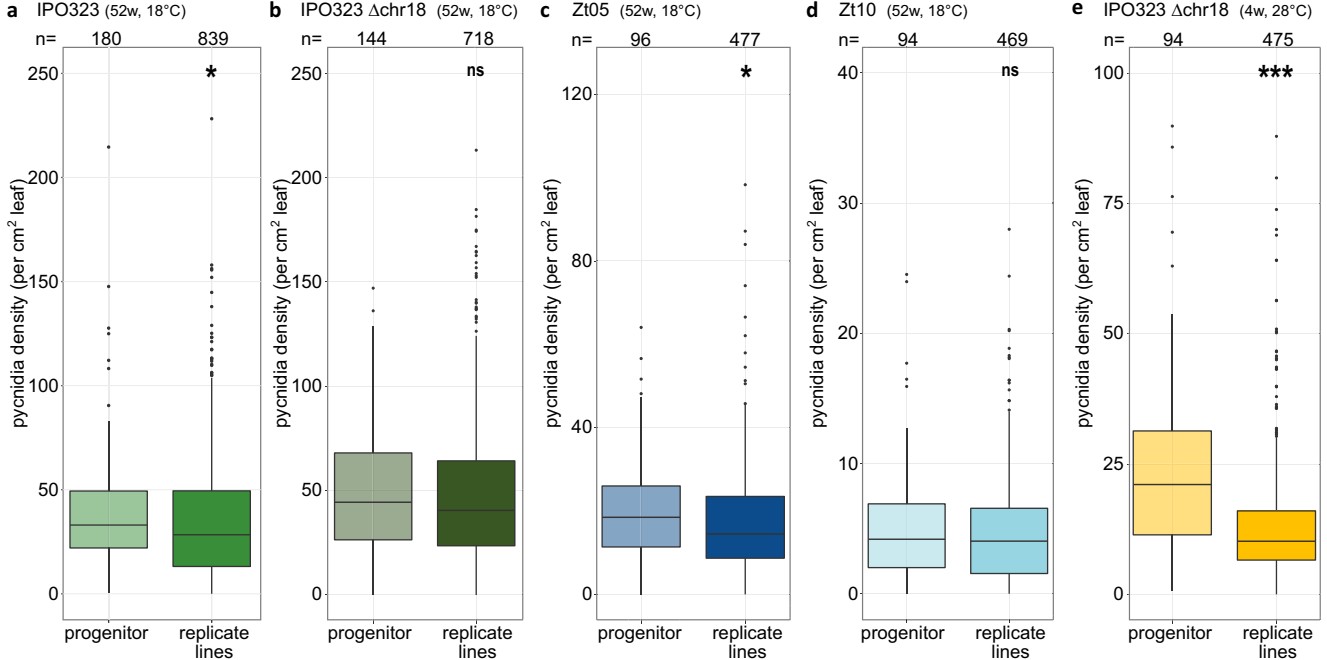

**Fig. 7 Accumulated mutations, on average, decrease the ability to produce pycnidia during infections in planta.** Effect of the accumulated mutations on the ability to produce pycnidia during infections of the wheat host for each of the indicated wild type strains, comparing the progenitor to pooled data for all the replicated MA lines ($n = 40$ independently evolved MA lines per strain). **a–e** Box-whiskers plot of the observed pycnidia density per $cm^2$ of leaf surface for the indicated strains comparing the progenitor line with the pooled data for the replicated MA lines for the indicated strains. Significant comparisons based on categorized $p$-values of Tukey's HSD on rank-based two-sided ANOVA are indicated by stars (*$p < 0.05$, **$p < 0.005$, ***$p < 0.0005$, ns = not significant at $\alpha = 0.05$). The exact $p$-values for of all pairwise comparisons are provided in Supplementary Data 1. Box plots depict center line, median; box limits, upper and lower quartiles; whiskers, 1.5× interquartile range; points, outliers.

increased mutation rates in accessory genomic compartments. Indeed, accessory chromosomes in other fungal pathogens such as *Alternaria alternata* and *F. oxysporum* encode such pathogenicity-related genes and drive a more rapid sequence evolution[31].

The distribution of specific histone marks has been correlated with mutation rates before[10,12,14]; however, experimental evidence for causality was so far missing. Here we addressed whether two histone modifications, H3K9me3 and H3K27me3, which are associated with heterochromatin, do correlate with differences in mutation rates in the wild type fungus and also how removal of these histone modifications impacts mutation rates. It is important to note that any change in the mutation rate observed upon the deletion of the responsible histone methyltransferases could be the result of one or more of the following scenarios: (i) the loss of histone modification directly affects the mutation rate locally in regions that are colocalizing with the histone modification in the wild type; (ii) the loss of histone modification results in a global change of cellular processes that affect the mutation rate globally; and (iii) the histone modification enzyme has another function that in turn affects the mutation rate (again this would presumably be on a more global scale). Hence, the differentiation between local and global effects should enable us to distinguish between these non-mutually exclusive hypotheses.

In the wild type, the presence of the histone mark H3K27me3 correlates with a higher mutation rate, which is similar to previous comparative genome studies including human cancer cells[10,12]. Removal of this histone modification reduced base substitution mutation rate in the Δ*kmt6* strain (no H3K27me3). This reduction of the mutation rate was not observed globally, but in the full-factorial analysis a significant difference to the wild type was observed in those regions colocalizing with H3K9me3, H3K27me3, and TEs (see Supplementary Data 1 for the results of

all statistical comparisons). Therefore, these results may suggest that H3K27me3 affects the mutation rate locally and thereby supports the role of H3K27me3 in directly promoting spontaneous mutations. We speculate that H3K27me3 promotes mutations via its effect on the chromatin structure mediated by DNA accessibility or nucleosome positioning[70–72]. Alternatively, H3K27me3 could, via its localization at the nuclear periphery[7], result in more exposure to mutagens or affect the DNA repair mechanism, or result in replication stress[57,73]. Indeed, we can infer DNA replication stress caused by H3K27me3 by the high frequency of chromosome losses that is not associated with an equally high frequency of chromosome duplications in the wild type. This low number of chromosome duplications supports that replication errors and not segregation errors are responsible for the losses of accessory chromosomes, which are enriched in H3K27me3, during mitosis. We therefore speculate that the promotion of mutations by H3K27me3 is mediated via its induction of replication stress. However, the presence of H3K27me3 does not influence the extent of larger structural variants in our study—which could indicate that these larger structural variants are DNA replication independent and could be caused by a different mechanism. DNA replication and replication timing has been recognized previously as a factor that could affect the mutation rate[10] but reported effects vary between studies[66]. We further suggest that the regions colocalized with H3K27me3 are associated with a later DNA replication[74]. However, we find the same mutation rate in regions that colocalized with H3K27me3 in the wild type for both the Δ*kmt6* strain and the wild type. Similar, albeit nonsignificant, changes in the mutation rate were also observed in regions not colocalizing with H3K27me3, which would indicate a global effect of H3K27me3 and/or KMT6. Such a global effect could be caused by different mechanisms and one possibility is that KMT6 affects the

mutation rate in an H3K27me3-independent manner—in analogy to observations in mammalian cells, where the respective homolog appears to be involved in DNA repair[75] or repair of DNA double-strand break[76], and can methylate non-histones such as transcription factors[77,78] independently of H3K27me3. Taken together, we conclude that H3K27me3 affects the mutation rate locally, whereas additional global effects of H3K27me3 and/ or KMT6 are at least indicated, requiring further experimental analysis in the future.

For regions of the genome colocalizing with H3K9me3, we do not see a significant effect on the base substitution rate in the wild type *Z. tritici* isolate, in contrast to the described effect of H3K9me3 in human cancer cell[10,12,73]. However, deleting *kmt1* resulted in a global increase of the base substitution rates and whole chromosome losses, as well as deletions and duplications —but a significant decrease in the rate of insertions. These global changes could be due to a direct effect, either via H3K9me3 on the metabolism or independently of H3K9me3, e.g., via non-histone substrates of KMT1, which have been reported for the mammalian homolog[78]. H3K9me3 is associated with TEs, is a hallmark of constitutive heterochromatin[11], and is known to be involved in genome maintenance[73]. Hence, its removal leads to transcriptional activation of TEs[58], which, in our study, correlates with the globally increased mutation rates and genome instability. Nevertheless, the increased genome instability appears not to result from more TE transposition, as the number of insertions is decreased in the Δ*kmt1* mutant lacking H3K9me3. Instead we hypothesize that the increase in genome instability may be caused by TE-associated illegitimate homologous recombination or deficient DNA repair as first observed and suggested by Möller et al.[58]. In the Δ*kmt1* deletion mutant, the largest structural duplications on chr1, chr8, chr9, and chr12 occurred in a region delimited by a TE on one side and the centromeric region on the other side—but many smaller duplications also exist within this region. Similar to Möller et al.[58], we therefore suggest that illegitimate recombination between TEs might have resulted in large duplications, spanning sequences from the respective TE to the centromere of the chromosome. Such events may be followed by additional rounds of mitotic recombination resulting in the smaller duplications that we also observe. In a previous study, we showed that H3K27me3 replaces H3K9me3 in the Δ*kmt1* mutant in *Z. tritici*[58]. Such a re-localization of the facultative hetero-chromatin mark H3K27me2/3 to sites colocalizing with H3K9me3 was also reported in a Δ*kmt1* mutant of *Neurospora crassa*, where it is associated with genotoxic stress[79,80]. The localization of H3K27me3 to sites previously colocalizing with H3K9me3 might increase the susceptibility of the affected regions to illegitimate homologous recombination as well as the mutation rate at these locations, possibly in a DNA replication-associated manner[58]. The lack of insertions, as well as the increase in chromosome losses without corresponding chromosome dupli-cations observed in this study support the hypothesis that DNA replication is involved in the effect of the H3K9me3 removal.

Stress-induced mutagenesis was shown in evolution experi-ments of *Arabidopsis thaliana*, *S. cerevisiae*, and *Caenorhabditis elegans* using harsh conditions that typically cause dramatic growth reduction[15,17,18]. Milder conditions that reduced growth by <50% resulted in a 3.6-fold increase in the mutation rate in *S. cerevisiae*[64]. Here we demonstrate a high increase in the mutation rate (~5-fold), a significant effect on the Ts/Tv ratio and an increase in larger deletions and duplications, as well as chromo-some losses and duplications all caused by a mild temperature stress that reduced the growth rate by <20%. Again, we do not see an increase in the rate of insertions, which would preclude TE mobilization as the main factor responsible for the increased mutations rates—in contrast to TE mobilization caused by

temperature stress as reported for the fungus *Cryptococcus deneoformans*[81]. On the leaf surface, *Z. tritici* frequently must experience 28 °C, even in temperate regions. We therefore spec-ulate that the increase in mutation rate at higher temperatures may confer a mechanism to rapidly generate new genomic variability upon which selection can act. Alternatively, the tem-perature stress may directly increase the mutation rate, e.g., by a cellular accumulation of compounds that damage DNA and therefore the increase of the mutation rate at higher temperatures would be coincidental rather than adaptive[82].

Finally, we show that DNA methylation impacts the genome stability in *Z. tritici*. Recently, we demonstrated that the *dim2* gene is responsible for 5mC[40]. This correlates with a considerably higher number of mutations in TEs[40]. Here we show that the presence of functional *dim2* also correlates with a lower number of structural variants and, in particular, a lower involvement of TEs in deletions. These data therefore support the hypothesis that the DIM2-dependent 5mC in *Z. tritici* could be a defense mechanism against TEs. TEs are known to be associated with structural or chromosomal variation via illegitimate homologous recombination that depend on sequence similarity in plants, fungi, and animals[83,84]. Hence, the reduced role of TEs in structural variation in Zt10 compared to Zt05 and IPO323 could be due to a lower sequence similarity among the TEs in Zt10 caused by a higher mutation rate in TEs. Indeed, we recently reported an overall lower content of TEs, as well as a lower GC content and distinct dinucleotide frequency in TEs in isolates containing a functional *dim2*, including Zt10, compared to iso-lates lacking a functional *dim2*[40]. Thereby, not only a lower number of TEs but also higher sequence dissimilarity between TEs in Zt10 may confer a lower rate of illegitimate homologous recombination, and then structural variants. Moreover, we report a lower expression from TEs in isolates with a functional *dim2*[40], which may result in a lower frequency of TE transposition. This may indicate that the *dim2*-associated high mutation rate reduces structural variation by both reducing the sequence similarity among the TEs and by reducing the TE-transposition process itself. The equilibrium GC content in Zt10 (5mC positive) was highly divergent from the observed GC content, an observation that could be explained by the restriction of this methylation and the C → T mutations to TEs[38], and therefore not applying to the whole genome. Nonetheless, it is still puzzling, how such a high rate of C → T transitions can occur without leaving a larger impact on the genome composition observed in field isolates.

In conclusion, this study determines the mutation rate and mutational spectrum in a pathogenic fungus, and demonstrates the importance of epigenetic modifications in shaping the mutation rate variation. The results underline that *Z. tritici*, with its complement of histone modifications and DNA methylation that are mostly absent from the experimental model species *S. cerevisiae* and *S. pombe*, provides an attractive eukaryote system to establish the underlying mechanisms of mutations and genome evolution.

## Methods

**Fungal material**. The Dutch isolate IPO323 is available from the Westerdijk Institute (Utrecht, The Netherlands) with the accession numbers CBS125943. Zt05 and Zt10 were isolated from Denmark and Iran, respectively[85]. IPO323Δchr18, a derivative of IPO323 that lost chromosome 18[39], IPO323Δchr18Δ*kmt1*, and IPO323Δchr18Δ*kmt6*[58] are available upon request.

**Fungal growth conditions**. All *Zymoseptoria spp.* isolates used in this study were cultivated at 18 °C in YMS (4 g/L yeast extract, 4 g/L malt, 4 g/L sucrose, and 20 g/L agar for plates) medium. Cultures for DNA extraction and plant infection experiments were inoculated directly from −80 °C glycerol stocks and grown on solid YMS medium at 18 °C (for plant infection experiments) or in liquid YMS medium at 18 °C at 200 r.p.m. for 5 to 7 days. Two replicated MA lines of Zt05 and two replicated MA

lines of Zt10 were excluded from the plant infection experiments, as they failed to produce a sufficient number of cells for inoculation. For a detailed description of the conditions during the MA experiment and the determination of the fungal growth rates (Supplementary Table 2), see Supplementary Methods.

**Phenotypic assays in planta**. Seedlings of the wheat cultivar Obelisk (Wiersum Plantbreeding BV, Winschoten, The Netherlands) were pre-germinated on wet sterile Whatman paper for 4 days under normal growth conditions (16 h at light intensity of ~200 μmol/m²/s and 8 h darkness in growth chambers at 20 °C with 90% humidity) followed by potting and further growth for additional 7 days. In planta assays were conducted as described in ref. [37]. All in planta phenotypic data are provided in Supplementary Data 3.

**Genome sequencing and data analysis**. For sequencing, DNA of 286 strains (6 progenitor and 280 evolved replicated MA lines) was isolated using a standard phenol–chloroform extraction protocol[37]. A summary of the sequencing data is provided in Supplementary Data 2. Library preparation and sequencing using an Illumina HiSeq2500 machine for the 286 strains were performed at the Max Planck Genome center, Cologne, Germany. Paired-end reads of 250 bp were mapped to the genome of the reference isolate IPO323 (Accession: GCA_000219625.1) and assemblies based on long-read sequencing of the two isolates Zt05 or Zt10[56,86]. Base substitutions, small INDELs, and structural variations were determined as described in the Supplementary Methods. A randomly selected subset of 65 mutations were verified using Sanger Sequencing (see Supplementary Data 3–5), and the sensitivity and specificity of identifying structural variations was determined using 820 simulated insertions and deletions (see Supplementary Data 6 and Supplementary Methods). Mutation rates were calculated per replicated MA line and the location of mutations in the following genomic compartments was determined for all IPO323-derived strains: gene models[42] (Supplementary Data 7), TEs[56] (Supplementary Data 8), and histone modifications (H3K4me2, H3K9me3, and H3K27me3)[39] (Supplementary Data 9–11). For each of the evolved replicated MA lines, the size of the genomic compartment was adjusted on sequencing coverage and the average size of the genomic compartments is given in Supplementary Table 3. For a detailed description of the DNA isolation, data analysis, and verification, see Supplementary Methods.

**Statistical analysis**. All statistical analyses were conducted in R (version R3.6.0) using the suite R Studio (1.2.1335). A summary of all statistical tests is given in Supplementary Data 1.

**Reporting summary**. Further information on research design is available in the Nature Research Reporting Summary linked to this article.

## Data availability

Sequencing reads have been deposited in the Sequence Read Archive and are available under the BioProject PRJNA614493 and PRJNA718981. The genome sequence of the reference isolate IPO323 used in this study is available under the accession GCA_000219625.1 at the European Nucleotide Archive. Supplementary Data 1–11 including the IPO323 gene annotations, regions of histone modification enrichment and TE annotations, and source data are available at https://doi.org/10.5281/zenodo.5413239. The assembled genomes of Zt05 and Zt10, and the respective annotations are available at https://zenodo.org/record/3820378.

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

## Acknowledgements

The study was funded by a personal grant to EHS from the State of Schleswig Holstein and the Max Planck Society, and in addition by a DFG-grant to M.H. (HA 9263/1-1). E.H.S. is moreover grateful for support from CIFAR. The funders had no role in study design, data collection and interpretation, or the decision to submit the work for publication. We thank Arne Traulsen for helpful support and Thorsten Reusch for his comments on an earlier version of this article.

## Author contributions

M.H. and E.H.S. designed the experiments. M.H. and J.K. performed the experiments. M.H., C.L., and A.F. analyzed the sequencing data. M.H. analyzed the results. M.H. and E.H.S. wrote the manuscript.

## Funding

## Competing interests

The authors declare no competing interests.
