## [Peer Review File · Nature Communications]

Epigenetic modifications modify the rate of spontaneous mutations in a pathogenic fungusReviewers' Comments:

Reviewer #1:

Remarks to the Author:

In this work, authors used mutation accumulation experiments to measure the mutation rate in the filamentous fungus *Zygomycetes tritici*, a wheat pathogen, during its mitotic growth. They focused on the roles of two types of histone modification (H3K27me3 and H3K9me3) and high temperature stress on the mutation rate by conducting the MA experiments in strains lacking genes encoding histone modification enzymes and in a high temperature (28C) environment, respectively. They report that H3K27me3 increases while H3K9me3 decreases the mutation rate, and that high temperature increases the mutation rate. Using a natural strain with a functional DIM2 gene, they found that cytosine methylation increases the mutation rate by 15 times in transposable elements (TEs) and decreases TE activity. While mutation rates have been measured using the MA approach in many microbes, the strength of the present study is the use of genotypes lacking two types of histone modification, allowing a direct assessment of their impact on mutation rate. I have one set of major comments all about the interpretation of the data and a few minor comments.

Major comments.

1. The authors correctly pointed out in the Introduction that observing different mutation rates at genomic regions with different histone modifications in previous studies of wild-types do not prove a causal relationship between histone modification and mutation rate. For example, the correlation between mutation rate and histone modification may be because both are influenced by a third factor such as the local genomic sequence. It is important to explicitly state this possibility, as it is relevant to the data interpretation (see below).
2. When the mutation rate is altered upon the deletion of a histone modification enzyme gene, there are several possibilities. First, loss of histone modification affects the mutation rate of local genomic regions normally marked by the histone modification. Second, loss of histone modification induces a change in cell physiology (or other cellular processes) such that the mutation rate is affected globally. Third, in addition to histone modification, the histone modification enzyme gene has other functions that affect mutation rate (presumably globally). It is important to explicitly list these probabilities because they affect the interpretation (see below).
3. Now let us try to interpret Fig. 5. In panel A, there are 7 sets of comparisons. In all 7 sets, we see that deleting *kmt1*, which removes H3K9me3, increases the mutation rate (by a similar amount). This means that deleting *kmt1* increases the mutation rate similarly across the genome, so its effect is global. This conclusion is actually very different from the previous correlation-based inference that H3K9me3 affects mutation rate locally at genomic sites marked with H3K9me3, and suggests that the previous observation of the correlation is non-causal, as hypothesized in the about first comment. The same is true for *kmt6* deletion; in all 7 sets of comparisons, deleting *kmt6* reduces the mutation rate by a similar amount. Thus, the present finding about the impact of histone modification (more precisely the impact of the histone modification enzyme genes) on mutation is actually a completely new discovery. Rather than confirming previous correlation-based conclusion, it shows that the impact of histone modification (enzyme genes) on the mutation rate is not through affecting local DNA accessibility, but likely through a global mechanism such as affecting cellular physiology or other cellular processes.

Minor comments.

1. Line 103. "Interestingly, populations of *Z. tritici* comprise an unusually high standing variation with 44% of all nonsynonymous substitutions estimated to be adaptive, which appears at odds with the basic tenet that most mutations in functionally relevant loci are assumed to confer a deleterious or slightly deleterious effect." The basic tenet is that most mutations, not most substitutions, are deleterious. So, either the sentence is wrong, or 44% is actually referring to mutations instead of substitutions. Please clarify.
2. Line 492. "On the leaf surface, *Z. tritici* frequently must experience 28°C, even in temperate regions. We therefore speculate that the increase in mutation rate at higher temperatures may confer

a mechanism to rapidly generate new genomic variability upon which selection can act." If these organisms frequently encounter 28C, they should evolve phenotypic plasticity such as gene regulation instead of waiting for new beneficial mutations every time the temperature rises to 28C. So this statement is likely false. It should be noted that the elevated mutagenesis under stress may not be adaptive (PMID: 23400102).

Reviewer #2:

Remarks to the Author:

In this manuscript, the authors assessed the impact of histone modifications associated with heterochromatin (H3K9me3 and H3K27me3), temperature stress (18° vs 28°) and cytosine methylation (5mC) on the genome-wide mutation rate and mutation spectra for the pathogenic plant fungus *Zygomycetozoria tricoli*. Spontaneous mutations in the genomes of the wildtype strain IPO323 were compared with deletion mutants lacking H3K9me3 or H3K27me3 chromatin marks. The effect of temperature stress was monitored in IPO323. Finally, the authors compare the impact of 5mC methylation using (unrelated) *Z. tricoli* field strains with or without a functional 5mC system. The experimental approach used mutation accumulation (MA) lines in which single colony passage through successive bottlenecks minimizes the impact of selection. In all cases except the high-temperature stress, 40 independent MA lines of each strain were passaged for 52 weeks; passage was only for 4 weeks at 28°. A quantitative assessment of mutation rates and types (base substitutions, indels and structural variants) is presented and the manuscript is generally well-written. That being said, there are suggestions below for streamlining the presentation and making figures less data dense. The authors provide evidence that 1) epigenetic modifications either increase (H3K27me3) or decrease (H3K9me3) the mutation rate, 2) temperature stress independently increases the mutation rate and 3) mutations are increased in specific genomic compartments, most often associated with accessory chromosomes and regions enriched for TEs and epigenetic modifications. Results are largely descriptive and are consistent with previous studies that have correlated histone modifications with the rates and types of genomic mutations. Although the mechanisms underlying the observed unclear, this study nevertheless offers important clues and lays the foundation for future studies. Below are comments that should be addressed by the authors.

Major comments:

1. The methylation data were derived using field isolates Zt05 (no 5mC) and Zt10 (5mC present), which are from distinct geographical regions. This is in contrast to the histone-modification MA lines, which used otherwise identical genomic backgrounds. Although a marked increase in the rate of base substitutions in TE-rich regions in Zt10 and a lower rate of structural variation (deletions, insertions and duplications) was found relative to the Zt05/IPO323 MA lines, one cannot draw strong conclusions from this analysis. The correct way to do this is to delete DIM2 in the Zt10 background and repeat the analysis. Otherwise, these data should be deleted. Deleting these data also would simplify the take-home messages.
2. Why were the MA lines at 28° propagated for only 4 weeks? Is it possible that exposure of *Z. tricoli* to a higher temperature stress (28°) results in a burst of mutations over a few generations instead of a constant rate of spontaneous genomic mutations per generation over time?
3. Use of the phrase "TE activity" throughout the manuscript is unclear. Does this mean TE mobilization or TE transcription? It seems that it may mean different things in different contexts.
4. The authors overstate findings in several places. For example, change last sentence of abstract to "environmental conditions modify... and alter its evolutionary trajectory."
5. The authors should remove throughout the use of "the first" to describe the findings, as this is often inaccurate. An increase in mutation rate in response to heat stress in the human pathogenic fungus

Cryptococcus has been reported (Gusa and Williams et al. PNAS 2020). In addition, a global analysis of mutations driving microevolution of a heterozygous diploid fungal pathogen has been published (Ene et al. PNAS 2018).

6. The IPO323 derivative missing chr 18 was used for the important comparisons. Why not just remove the IPO323 data and simplify the comparisons?

7. The first time it is mentioned, the authors need to state that H3K4me2 is a repressive mark. The authors might want to consider excluding these data as it is peripheral to the main points.

8. If the authors are going to devote a section to fitness effects, then at least some of the data should be included in the main text (all are in Figs S8-S11). In looking at these data and the inherent variability between different MA lines, it seems difficult to come to any sweeping conclusions.

Minor comments:

Fig 1B – the strain names against some of the dark backgrounds used are hard to read. Why not just used the lighter colors here as well?

line 167 – should be Zt05

lines 324-6 are not supported by the data shown.

line 52: mechanism (mechanisms)

line 101: exist (exists)

line 113: theses (these)

line 305: opposite (opposite), loses (loss)

line 263: wildtpye (wildtype)

line 474: transcriptionally (transcriptional)

line 502: correlate (correlates)

Reviewer #3:

Remarks to the Author:

Habig and colleagues carried out mutation accumulation experiments with the wheat pathogen *Zymoseptoria tritici* to examine the effects of epigenetic modifications and temperature. Strains were grown for 52 weeks with weekly bottlenecks and whole genome sequencing was performed to analyze the frequency of base substitutions, insertions/deletions, and chromosome loss/gains. Multiple field isolates were analyzed, including one isolate with a functional DNA methylation system, and mutant strains that lack the histone H3 lysine-9 and lysine-27 methyltransferases were examined (KMT1 and KMT6, respectively). In addition, a temperature stress condition was analyzed for a wild type isolate. Based on genome re-sequencing, the authors propose that: (1) mutation rates differ on core and accessory chromosomes, (2) H3K9 methylation represses mutation, (2) H3K27me3 stimulates mutation, (3) DNA methylation limits TE-associated genome instability, and (4) temperature stress elevates mutation rate. This work extends prior work published by the Stukenbrock lab, providing a more quantitative analysis of some previously reported phenotypes of *Z. tritici* strains (Moller et al. 2019, PLoS Genetics, Moller et al. 2021, PLoS Genetics). The paper is well written, the analysis is rigorous, and the data are clearly presented in most cases. Some weaknesses should be addressed prior to publication. 1) The conclusion that epigenetic modifications impact stability is consistent with the data but other possibilities are not discussed (see #1 below). 2) KMT1 and KMT6 have been previously linked to genome stability in multiple fungi. The current work provides further support for this role, but does not provide new mechanistic insights. For example, the role of repeated DNAs and % identify of DNA repeat are not explored in the current manuscript.

1) One major conclusion is that epigenetic modifications impact mutation rate. While deletion mutants of *kmt1* and *kmt6* are analyzed, it should be noted that homologs of KMT1 and KMT6 methylate

histones and non-histone substrates in other organisms. It is possible that a non-histone substrate is critical for the genome stability phenotypes reported here. While this seems unlikely for *kmt1*, in Figure 5B the mutation rate in sequences associated with H3K27me3 alone is similar in wild type and the *kmt6* mutant. Because PRC2 is linked to double strand break repair in mammals (e.g. Cambell et al. 2013, and others), the possibility that a non-histone substrate of PRC2 impacts genome stability should be discussed and the conclusion that H3K27me3 impacts mutation rates should be rephrased to reflect alternate hypotheses that are consistent with the data. A direct role for H3K9me3 in genome stability has been demonstrated in other organisms and is therefore likely but not proven based on the experiments here.

2) Mechanism - The defects reported in wild type and for the *kmt1* mutant and for 5mC-proficient strains may be consistent with illegitimate recombination between repeated DNA as an important source of genome instability.

Figure S4 – Is this figure showing multiple independently evolved lines? If so, *kmt1* strains frequently duplicate similar regions. Is there a role for repeated DNAs in recurring or highly similar rearrangement events? Perhaps additional discussion of these recurring rearrangements could provide mechanistic clues.

Figure 6 – Does increased mutation rate in ZT10 correlate with reduced sequence similarity between TEs? Is it possible that fewer substrates for Illegitimate homologous recombination exist in this strain?

Other comments

Figure 4D and line 455-460 – is it possible that duplication of accessory chromosomes negatively impacts fitness more than loss of accessory chromosomes? Was the fitness tested (fig S8) in strains with chromosome duplications? If so, how did the fitness of these strains compare to strains with accessory chromosome losses?

Line 35: “detailed insights are central”

Line 194 – “some regions with a particular combination of features was small” - Interpreting these factorial data is difficult without knowing the total size of each compartment. Including the size in kb for each category in the figure legend or supplementary table would be helpful.

Line 214.... “but not H3K9me3 were associated with a significantly higher base substitution rate.” - I assume that most H3K9me3 is associated with TEs, which has the highest rate of base substitutions. Is it possible that this substitution rate is not statistically higher than control regions because it comprises a relatively small fraction of the genome? If so, the statement is a bit misleading. As mentioned above, it would be helpful to have a supplementary table that includes the size in base pairs of each category.

The statement on line 218-219 is redundant with the statement on line 214.

Line 299 - “insertions that did occur were specific to regions otherwise co-localizing with the H3K9me3 (Fig 5E)” - the location of insertions is not shown in Fig 5E.

Line 472 – chromosome losses

Line 481 – relocalization of H3K27me2/3 in *kmt1* mutants and its association with fungal genome instability was first shown in *Neurospora* (Basenko et al. 2015 and Jamieson et al. 2016). These papers should be cited.

Rebuttal letter: “Epigenetic modifications modify the rate of spontaneous mutations in a pathogenic fungus”

Detailed reply to the editor and the reviewers’ comments.

Thank you for the constructive critique and comments to our manuscript. We have addressed each of the points raised by the reviewers and provide below our replies and a summary of changes to the manuscript. Please note that the reviewer comments are given in black, and our replies in blue. Please also see the revised main text, in which we have tracked the introduced changes. All line numbers given below refer to these revised files.

Reviewer #1 (Remarks to the Author):

In this work, authors used mutation accumulation experiments to measure the mutation rate in the filamentous fungus *Zymoseptoria tritici*, a wheat pathogen, during its mitotic growth. They focused on the roles of two types of histone modification (H3K27me3 and H3K9me3) and high temperature stress on the mutation rate by conducting the MA experiments in strains lacking genes encoding histone modification enzymes and in a high temperature (28C) environment, respectively. They report that H3K27me3 increases while H3K9me3 decreases the mutation rate, and that high temperature increases the mutation rate. Using a natural strain with a functional DIM2 gene, they found that cytosine methylation increases the mutation rate by 15 times in transposable elements (TEs) and decreases TE activity. While mutation rates have been measured using the MA approach in many microbes, the strength of the present study is the use of genotypes lacking two types of histone modification, allowing a direct assessment of their impact on mutation rate. I have one set of major comments all about the interpretation of the data and a few minor comments.

Major comments.

1. The authors correctly pointed out in the Introduction that observing different mutation rates at genomic regions with different histone modifications in previous studies of wild-types do not prove a causal relationship between histone modification and mutation rate. For example, the correlation between mutation rate and histone modification may be because both are influenced by a third factor such as the local genomic sequence. It is important to explicitly state this possibility, as it is relevant to the data interpretation (see below).

Our response: We thank the reviewer for this comment. In the revised manuscript, we underline the possibility of other direct and indirect factors on the mutation rate. This hypothesis is now introduced in the introduction and discussed in a new additional paragraph in the discussion (please see below). (line 48-51, 474-483)

2. When the mutation rate is altered upon the deletion of a histone modification enzyme gene, there are several possibilities. First, loss of histone modification affects the mutation rate of local genomic regions normally marked by the histone modification. Second, loss of histone modification induces a change in cell physiology (or other cellular processes) such that the mutation rate is affected globally. Third, in addition to histone modification, the histone modification enzyme gene

has other functions that affect mutation rate (presumably globally). It is important to explicitly list these probabilities because they affect the interpretation (see below).

Our response: We have included a paragraph in the discussion that explicitly lists these possibilities and discuss our results based on such additional mechanisms and scenarios (please see below). (line 474-483)

3. Now let us try to interpret Fig. 5. In panel A, there are 7 sets of comparisons. In all 7 sets, we see that deleting *kmt1*, which removes H3K9me3, increases the mutation rate (by a similar amount). This means that deleting *kmt1* increases the mutation rate similarly across the genome, so its effect is global. This conclusion is actually very different from the previous correlation-based inference that H3K9me3 affects mutation rate locally at genomic sites marked with H3K9me3, and suggests that the previous observation of the correlation is non-causal, as hypothesized in the about first comment. The same is true for *kmt6* deletion; in all 7 sets of comparisons, deleting *kmt6* reduces the mutation rate by a similar amount. Thus, the present finding about the impact of histone modification (more precisely the impact of the histone modification enzyme genes) on mutation is actually a completely new discovery. Rather than confirming previous correlation-based conclusion, it shows that the impact of histone modification (enzyme genes) on the mutation rate is not through affecting local DNA accessibility, but likely through a global mechanism such as affecting cellular physiology or other cellular processes.

Our response: We agree that a global mechanism could be responsible for the observed pattern in the effect of the histone modifications on the mutation rate and have included a section in the discussion on this possibility which also states the three possibilities mentioned above. We also agree that in figure 5a, showing the seven comparisons, the effect of removal of the histone modifications is similar between the different, considered genomic compartments. However, since TEs, H3K9me3 and H3K27me3 do overlap we included the full factorial analysis (in Fig. 5b) in which the effect of the removal of the histone modifications is observed for each possible combination of the three marks. Again, a similar increase caused by the removal of H3K9me3 was observed for any of the combinations – which again might indicate a global role of this histone modification – which we have now addressed more thoroughly in the discussion. This is different for the removal of H3K27me3. Here, in only one of the combinations of genomic marks a significantly reduced mutation rate compared to the wildtype was observed, namely in regions that were associated with TEs, H3K9me3 and H3K27me3. This result is more in line with a local effect of the Histone modifications – even if it does not exclude a global effect. We have included both lines of argument in the discussion. Since at this stage it is difficult to further disentangle local from global effects we have included both hypotheses for the effect of H3K27me3 in the discussion. (line 474-483, 488-493, 513-520)

Minor comments.

1. Line 103. “Interestingly, populations of *Z. tritici* comprise an unusually high standing variation with 44% of all nonsynonymous substitutions estimated to be adaptive, which appears at odds with the basic tenet that most mutations in functionally relevant loci are assumed to confer a deleterious or slightly deleterious effect.” The basic tenet is that most mutations, not most substitutions, are deleterious. So, either the sentence is wrong, or 44% is actually referring to mutations instead of substitutions. Please clarify.

Our response: Indeed, we meant mutations and not substitutions. We have corrected our mistake in the manuscript accordingly. (line 106, 116, 395)

2. Line 492. “On the leaf surface, *Z. tritici* frequently must experience 28°C, even in temperate regions. We therefore speculate that the increase in mutation rate at higher temperatures may confer a mechanism to rapidly generate new genomic variability upon which selection can act.” If these organisms frequently encounter 28C, they should evolve phenotypic plasticity such as gene regulation instead of waiting for new beneficial mutations every time the temperature rises to 28C. So this statement is likely false. It should be noted that the elevated mutagenesis under stress may not be adaptive (PMID: 23400102).

Our response: We have adapted this section of the text and included a statement that the increased mutation rate at higher temperatures could be coincidental and not adaptive. (line 535-571)

Reviewer #2 (Remarks to the Author):

In this manuscript, the authors assessed the impact of histone modifications associated with heterochromatin (H3K9me3 and H3K27me3), temperature stress (18° vs 28°) and cytosine methylation (5mC) on the genome-wide mutation rate and mutation spectra for the pathogenic plant fungus *Zymoseptoria tritici*. Spontaneous mutations in the genomes of the wildtype strain IPO323 were compared with deletion mutants lacking H3K9me3 or H3K27me3 chromatin marks. The effect of temperature stress was monitored in IPO323. Finally, the authors compare the impact of 5mC methylation using (unrelated) *Z. tritici* field strains with or without a functional 5mC system. The experimental approach used mutation accumulation (MA) lines in which single colony passage through successive bottlenecks minimizes the impact of selection. In all cases except the high-temperature stress, 40 independent MA lines of each strain were passaged for 52 weeks; passage was only for 4 weeks at 28°. A quantitative assessment of mutation rates and types (base substitutions, indels and structural variants) is presented and the manuscript is generally well-written. That being said, there are suggestions below for streamlining the presentation and making figures less data dense. The authors provide evidence that 1) epigenetic modifications either increase (H3K27me3) or decrease (H3K9me3) the mutation rate, 2) temperature stress independently increases the mutation rate and 3) mutations are increased in specific genomic compartments, most often associated with accessory chromosomes and regions enriched for TEs and epigenetic modifications. Results are largely descriptive and are consistent with previous studies that have correlated histone modifications with the rates and types of genomic mutations. Although the mechanisms underlying the observed unclear, this study nevertheless offers important clues and lays the foundation for future studies. Below are comments that should be addressed by the authors.

Major comments:

1. The methylation data were derived using field isolates Zt05 (no 5mC) and Zt10 (5mC present), which are from distinct geographical regions. This is in contrast to the histone-modification MA lines, which used otherwise identical genomic backgrounds. Although a marked increase in the rate of base substitutions in TE-rich regions in Zt10 and a lower rate of structural variation (deletions, insertions and duplications) was found relative to the Zt05/IPO323 MA lines, one cannot draw strong conclusions from this analysis. The correct way to do this is to delete DIM2 in the Zt10 background and repeat the analysis. Otherwise, these data should be deleted. Deleting these data also would simplify the take-home messages.

Our response: We thank the reviewer for the comment and agree that the correct way to do this would be to use a deletion mutant of *dim2* in the Zt10 background and in addition, ideally IPO323 (or Zt05) with the functional copy of *dim2* (originated from Zt10) and repeat the mutation accumulation experiment with these strains.

In a previous study we have investigated the role of DNA methylation on genome evolution in *Z. tritici* (Möller et al. 2021). Herein, we present data from genetically manipulated strains of *Z. tritici* with and without *dim2*. We generated a Zt10 Δ *dim2* deletion mutant as well as a IPO323 Δ chr18::*dim2* (Möller et al. 2021), and show that the deletion of *dim2* from Zt10 resulted in markedly reduced methylations within TEs – to a level very similar to IPO323 Δ chr18 lacking a functional *dim2*. Also, for IPO323 Δ chr18::*dim2* a marked increase in the level of cytosine methylation in TEs occurred – to a level very similar to Zt10 (containing a functional *dim2*). Based on these results we concluded in our

previous study that *dim2* is responsible for the methylation in TEs in *Z. tritici*, and were able to correlate 5mC to a higher number of mutations in TEs and hypothesized that this might be a defence mechanism against TE mobilization (Möller et al. 2021) but could not include any supportive data to this hypothesis.

Our new analysis goes beyond our previous work, as it allows us to determine the exact mutation rates and correlate the presence of a functional *dim2* to a lower level of structural variation and involvement of TEs in deletions – thereby further supporting the hypothesis that 5mC might be used as a defence mechanism against TE mobilization. These are new insights, which we find important to report. At the same time, we agree with the reviewer that the current data still does not allow us to prove causality. Therefore, in response to the reviewer's concerns, we have carefully re-worded our conclusions in order to highlight that the data is generally consistent with the hypothesis that 5mC might be a defence mechanism against TE mobilization. (line 572-577)

We still consider this extensive data of mutation accumulation highly relevant in the context of the present manuscript, because the data provides new insights into the determinants of genome evolution in a fungal pathogen. Therefore, we decided to maintain the *dim2* data and analyses within this manuscript.

2. Why were the MA lines at 28° propagated for only 4 weeks? Is it possible that exposure of *Z. tritici* to a higher temperature stress (28°) results in a burst of mutations over a few generations instead of a constant rate of spontaneous genomic mutations per generation over time?

Our response: We have started the higher temperature treatment at a later stage, after recognizing that temperature could have a pronounced effect – which then only allowed us to continue the experiment for four weeks. We believe, that the very nature of the mutation accumulation experiment is ideally suited to not only compare different treatments but also different length of experiments by normalising with the number of cell divisions, as we have done here and as it has been done in other key studies based on experimental evolution or mutation accumulation (e.g. Lynch et al. 2016; Nguyen et al. 2020; Liu and Zhang 2019). Moreover, mutation accumulation experiments maximize drift and minimize selection. Therefore, any adaptation (e.g. to different environmental conditions or differences in mutation rates) should not occur. Conceptually, there is no difference in the selection regime at different time-points of the course of the mutation accumulation experiment. Hence, we believe that a burst of mutations over few generations followed by a lower rate of mutations in the subsequent cell generations is unlikely. In response to the reviewer's comment, we now briefly explain these points in the revised methods (Text S1) and results section. (line 151-153)

Moreover - in order to assess if there is a time-dependency of mutations we have analysed the loss of accessory chromosomes during the course of the 52-weeks for IPO323-derived strains with all replicates in the mutation accumulation experiment. We detected the presence of the accessory chromosomes using two PCR-based markers for each accessory chromosome at 13, 26, 39 or 52 weeks of propagation. For all four different IPO323-derived strains a near linear relationship between the number of lost chromosomes and the propagation duration was observed (see Figure A below). In addition, we compared the observed rate of chromosome losses with the expected rate of losses assuming a constant rate over 52 weeks. For none of the time-points a significant deviation of the observed from the expected rate of chromosome losses was detectable (see Table A below). Therefore, (large) changes/mutations did occur at a constant rate during the course of the mutation accumulation experiment and we assume that this is also true for (smaller) base substitution mutations. We therefore believe that there were constant mutation rates at 18° C and assume that

this was also true at 28°C and therefore believe that our comparison between the different treatments is valid although the treatment lengths were not identical.

Figure A. Losses of accessory chromosomes over the course of the 52 weeks of propagation during the mutation accumulation experiment for the four IPO323 derived strains (each with 40 evolved, replicated lines). A chromosome was considered lost, if it failed to produce two specific PCR products located at the opposite subtelomeric regions of the chromosome. Total number of chromosomes summed up for all 40 replicates of each strain after 0, 13, 26, 39 and 52 weeks of propagation are depicted.

Table A. Observed and simulated number of chromosome losses during the mutation accumulation experiment.

Strain	Type of data	Replicates	# accessory chromosomes per genome	# accessory chromosomes at start	# lost accessory chromosomes after:			
					13 weeks	26 weeks	39 weeks	52 weeks
IPO323	observed data	40	8	320	7	11	17	25
	simulated data*	40	8	320	6	13	19	25
	p-value**	-	-	-	1	0.836	0.864	1
IPO323 Δ chr18	observed data	40	7	280	11	15	17	21
	simulated data*	40	7	280	5	11	16	21
	p-value**	-	-	-	0.204	0.548	1	1
IPO323 Δ chr18 Δ kmt1	observed data	40	7	280	15	23	34	52
	simulated data*	40	7	280	13	26	39	52
	p-value**	-	-	-	0.847	0.765	0.616	1
IPO323 Δ chr18 Δ kmt6	observed data	40	7	280	0	0	3	5
	simulated data*	40	7	280	1	3	4	5
	p-value**	-	-	-	1	0.249	1	1

* assuming constant loss-rate over 52 weeks

**Fisher's exact test (observed vs. simulated)

3. Use of the phrase “TE activity” throughout the manuscript is unclear. Does this mean TE mobilization or TE transcription? It seems that it may mean different things in different contexts.

Our response: It means TE mobilization. We have clarified this in the text.

4. The authors overstate findings in several places. For example, change last sentence of abstract to “environmental conditions modify... and alter its evolutionary trajectory.”

Our response: We have carefully modified our wording throughout the manuscript and in the title.

5. The authors should remove throughout the use of “the first” to describe the findings, as this is often inaccurate. An increase in mutation rate in response to heat stress in the human pathogenic fungus *Cryptococcus* has been reported (Gusa and Williams et al. PNAS 2020). In addition, a global analysis of mutations driving microevolution of a heterozygous diploid fungal pathogen has been published (Ene et al. PNAS 2018).

Our response: We thank the reviewer for this comment. We have rephrased our mentioning of “the first” and have now quoted the suggested references in the introduction and discussion. (line 121, 433, 595)

6. The IPO323 derivative missing chr 18 was used for the important comparisons. Why not just remove the IPO323 data and simplify the comparisons?

Our response: We understand the request to simplify our presentation of *Z. tritici* strains used in our study. However, although all comparisons of IPO323 Δ chr18 Δ km1 and IPO323 Δ chr18 Δ km6 strains are to IPO323 Δ chr18 and therefore do not include any data of the IPO323, we also focus on the correlation between histone modifications and the mutation rate in the wildtype. Here, including data from both IPO323 & IPO323 Δ chr18 for which information of the histone modifications is available further increases the statistical power and resolution of comparisons between different genomic regions within the independent replicates. In addition, we also include comparison between isolates IPO323, Zt05 and Zt10 and consider it important to use the IPO323 wildtype for these comparisons. In response to the reviewer’s concerns, we now briefly explain our reasoning in the revised methods section (Text S1).

7. The first time it is mentioned, the authors need to state that H3K4me2 is a repressive mark. The authors might want to consider excluding these data as it is peripheral to the main points.

Our response: We have include a statement that in fungi H3K4me2 is considered to be associated with transcriptionally active euchromatin (Freitag 2017), the first time it is mentioned. In previous studies we have furthermore shown that H3K4me2 is associated with transcriptionally active regions in the genome of *Z. tritici* (Schotanus et al. 2015; Feurtey et al. 2020). Although the H3K4me2 correlation is peripheral to the main points we decided to include these data as its correlation with lower mutation rates serves as a contrast to the correlations observed for H3K9me3 and, in particular, H3K27me3. (line 193-194)

8. If the authors are going to devote a section to fitness effects, then at least some of the data should be included in the main text (all are in Figs S8-S11). In looking at these data and the inherent variability between different MA lines, it seems difficult to come to any sweeping conclusions.

Our response: We have moved the figure S8 to the main text. Since this is the first time the effect of random mutations on the fitness of a plant pathogen is measured so extensively we believe that the inclusion of these data adds a valuable aspect to the study, and we have therefore decided to include it in the manuscript. This aspect is now highlighted in the results section. (line 418-427)

Minor comments:

Fig 1B – the strain names against some of the dark backgrounds used are hard to read. Why not just used the lighter colors here as well?

Our response: We have adjusted the colour of the strain names in Fig 1B.

line 167 – should be Zt05

Our response: We have corrected this mistake. (line 173)

lines 324-6 are not supported by the data shown.

Our response: We have removed this sentence. (line 336-338)

line 52: mechanism (mechanisms)

Corrected. (line 54)

line 101: exist (exists)

Corrected. (line 103)

line 113: theses (these)

Corrected. (line 117)

line 305: opposite (opposite), loses (loss)

Corrected. (line 316-317)

line 263: wildtpye (wildtype)

Corrected. (line274)

line 474: transcriptionally (transcriptional)

Corrected. (line 530)

line 502: correlate (correlates)

Corrected. (line 575)

Reviewer #3 (Remarks to the Author):

Habig and colleagues carried out mutation accumulation experiments with the wheat pathogen *Zymoseptoria tritici* to examine the effects of epigenetic modifications and temperature. Strains were grown for 52 weeks with weekly bottlenecks and whole genome sequencing was performed to analyze the frequency of base substitutions, insertions/deletions, and chromosome loss/gains. Multiple field isolates were analyzed, including one isolate with a functional DNA methylation system, and mutant strains that lack the histone H3 lysine-9 and lysine-27 methyltransferases were examined (KMT1 and KMT6, respectively). In addition, a temperature stress condition was analyzed for a wild type isolate. Based on genome re-sequencing, the authors propose that: (1) mutation rates differ on core and accessory chromosomes, (2) H3K9 methylation represses mutation, (2) H3K27me3 stimulates mutation, (3) DNA methylation limits TE-associated genome instability, and (4) temperature stress elevates mutation rate.

This work extends prior work published by the Stukenbrock lab, providing a more quantitative analysis of some previously reported phenotypes of *Z. tritici* strains (Moller et al. 2019, PLoS Genetics, Moller et al. 2021, PLoS Genetics). The paper is well written, the analysis is rigorous, and the data are clearly presented in most cases. Some weaknesses should be addressed prior to publication. 1) The conclusion that epigenetic modifications impact stability is consistent with the data but other possibilities are not discussed (see #1 below). 2) KMT1 and KMT6 have been previously linked to genome stability in multiple fungi. The current work provides further support for this role, but does not provide new mechanistic insights. For example, the role of repeated DNAs and % identify of DNA repeat are not explored in the current manuscript.

1) One major conclusion is that epigenetic modifications impact mutation rate. While deletion mutants of *kmt1* and *kmt6* are analyzed, it should be noted that homologs of KMT1 and KMT6 methylate histones and non-histone substrates in other organisms. It is possible that a non-histone substrate is critical for the genome stability phenotypes reported here. While this seems unlikely for *kmt1*, in Figure 5B the mutation rate in sequences associated with H3K27me3 alone is similar in wild type and the *kmt6* mutant. Because PRC2 is linked to double strand break repair in mammals (e.g. Cambell et al. 2013, and others), the possibility that a non-histone substrate of PRC2 impacts genome stability should be discussed and the conclusion that H3K27me3 impacts mutation rates should be rephrased to reflect alternate hypotheses that are consistent with the data. A direct role for H3K9me3 in genome stability has been demonstrated in other organisms and is therefore likely but not proven based on the experiments here.

Our response: We thank the reviewer for this comment. We have now included a general paragraph outlining the alternative hypotheses in the discussion. We further explain these alternative hypotheses while discussing the results for both the *kmt1* and the *kmt6* deletion strains. (line 474-483)

2) Mechanism - The defects reported in wild type and for the *kmt1* mutant and for 5mC-proficient strains may be consistent with illegitimate recombination between repeated DNA as an important source of genome instability.

Our response: We have now included recombination between repeated DNA as a possible mechanism in our discussion of the results (please see below).

Figure S4 – Is this figure showing multiple independently evolved lines? If so, *kmt1* strains frequently duplicate similar regions. Is there a role for repeated DNAs in recurring or highly similar rearrangement events? Perhaps additional discussion of these recurring rearrangements could provide mechanistic clues.

Our response: Indeed, the figure S4 shows multiple independently evolved lines and specifically multiple duplications in the IPO323 Δ Chr18 Δ *kmt1* within defined regions of chr1, chr8, chr9 and chr12. Although the length of the individual duplications varies within these regions, on closer inspection the longest duplications within each of the chromosomes are in all cases limited by an annotated transposable element on one side and the centromeric region on the other side. However, it is important to note that many large duplications exist that are only spanning part of the above-mentioned regions in chr1, chr8, chr9 and chr12.

We already identified a similar pattern in a previous study where the histone mutants were first presented (Möller et al. 2019). The removal of H3K9me3 in the Δ *kmt1* mutant seems to have triggered this instability – possibly linked to the enrichment of H3K27me3 on sites covered by H3K9me3 in the wildtype (Möller et al. 2019). Therefore, in our previous study, we hypothesised that this pattern is due to a TE-associated instability caused by H3K9me3 loss or the increase of H3K27me3 at former H3K9me3 sites, followed by continuous rearrangements possibly by mitotic recombination or deficiency in DNA repair (Möller et al. 2019). We have included this possible explanation with an emphasis on the possible role of illegitimate mitotic recombination in the revised discussion. (line 533-552)

Figure 6 – Does increased mutation rate in ZT10 correlate with reduced sequence similarity between TEs? Is it possible that fewer substrates for Illegitimate homologous recombination exist in this strain?

Our response : In a recent study we compared the sequence composition of the TEs in isolates containing a functional (including Zt10) or isolates lacking functional *dim2* (including IPO323 Δ chr18). We found an overall lower TE content, and a lower GC content and differences in dinucleotide frequency in TE regions in Zt10. These patterns correlate with the occurrence of DNA methylation almost exclusively in TEs (Möller et al. 2021). This would support the hypothesis that there could be a lower number of TE-substrates for illegitimate recombination in Zt10 compared to IPO323 and Zt05. In the same study we also observed a lower number of TEs showing expression in strains with a functional *dim2*. This would furthermore support that the accumulated mutations affected the expression of TE-associated genes and thereby assumingly the transposition process of TEs. We can currently not distinguish between these two not mutually exclusive hypotheses. In response to the reviewer's comment, we have therefore included both in the revised discussion. (line 577-590)

Other comments

Figure 4D and line 455-460 – is it possible that duplication of accessory chromosomes negatively impacts fitness more than loss of accessory chromosomes? Was the fitness tested (fig S8) in strains with chromosome duplications? If so, how did the fitness of these strains compare to strains with accessory chromosome losses?

Our response: To answer this question we have re-analysed the *in vitro* growth characteristics of the evolved replicated lines of IPO323 Δ chr18 (4w, 28°C) for which growth curves had been established and reported in the manuscript. We grouped the data according to whether the replicated lines had lost at least one chromosome (17 replicated lines) or duplicated at least one chromosome (9 replicated lines) and compared the growth characteristics to evolved replicates that had not lost nor duplicated chromosomes (10 replicated lines). Neither the carrying capacity nor the maximum growth rate

(μ_{\max}) significantly differed between these groups of replicated lines (see Fig. B of this rebuttal letter). Hence, there seems to be no evidence to support a difference in fitness *in vitro* between replicated lines that had lost or duplicated chromosomes. We therefore believe that the observed lower frequency of duplicated chromosomes (compared to the frequency of chromosome loss) supports that replication and not segregation errors are responsible for the high loss rates – and have therefore kept this statement in line 455-460. (now line 490-494)

Figure B. Comparison of *in vitro* growth of the 40 evolved replicated lines of IPO323 Δ chr18 that had been grown for 4 weeks at 28°C as characterised by the A) the maximum carrying capacity or B) the maximum growth rate (μ_{\max}), categorized according to whether chromosomes had been lost or duplicated. In addition, the growth characteristics of the progenitor is depicted. The results of statistical comparisons compared to the group of replicated lines that did neither lose nor duplicate chromosomes are depicted. Categorized FDR-corrected p-values of Wilcoxon rank sum tests are shown (*: $p < 0.05$, **: $p < 0.005$, ***: $p < 0.0005$).

Line 35: “detailed insights are central”

Corrected. (line 35)

Line 194 – “some regions with a particular combination of features was small” - Interpreting these factorial data is difficult without knowing the total size of each compartment. Including the size in kb for each category in the figure legend or supplementary table would be helpful.

Our response: We have included a summary table with the average size of the regions for all IPO323-derived strains for all 40 replicates and all environmental conditions in Table S3

Line 214.... “but not H3K9me3 were associated with a significantly higher base substitution rate.” - I assume that most H3K9me3 is associated with TEs, which has the highest rate of base substitutions. Is it possible that this substitution rate is not statistically higher than control regions because it comprises a relatively small fraction of the genome? If so, the statement is a bit misleading. As mentioned above, it would be helpful to have a supplementary table that includes the size in base pairs of each category.

Our response. We have included the supplementary table S3 that shows the average size of each of the compartments. The average size of the H3K9me3 co-localizing region, that is not TE and not co-localizing with H3K27me3 is 1.4 Mb. In this region a total of 34 SNPs were identified (IPO323 and IPO323 Δ chr18). We consider this number of observed SNPs large enough to be only marginally affected by stochasticity and therefore the statistical power to be high enough to detect differences

if they existed. Therefore, we consider the mentioned statement to be consistent with the data. These aspects are now mentioned and explained in the revised methods and materials (Text S1).

The statement on line 218-219 is redundant with the statement on line 214.

Our response: We have deleted this sentence. (line 223-225)

Line 299 - "insertions that did occur were specific to regions otherwise co-localizing with the H3K9me3 (Fig 5E)" - the location of insertions is not shown in Fig 5E.

Our response: We have corrected this mistake and referenced the correct figure Fig S6B. We have also corrected the sentence to describe the locations of the insertions that occurred in the mutants lacking H3K9me3. (line 308)

Line 472 – chromosome losses

Corrected. (line 516)

Line 481 – relocalization of H3K27me2/3 in kmt1 mutants and its association with fungal genome instability was first shown in Neurospora (Basenko et al. 2015 and Jamieson et al. 2016). These papers should be cited.

Our response. We thank the reviewer for these suggestions. We were unaware of the listed references and have included them in our discussion. (line 537)

Publication bibliography

Feurtey, Alice; Lorrain, Cécile; Croll, Daniel; Eschenbrenner, Christoph; Freitag, Michael; Habig, Michael et al. (2020): Genome compartmentalization predates species divergence in the plant pathogen genus *Zyoseptoria*. In *BMC genomics* 21 (1), pp. 1–15.

Freitag, Michael (2017): Histone Methylation by SET Domain Proteins in Fungi. In *Annual review of microbiology* 71, pp. 413–439. DOI: 10.1146/annurev-micro-102215-095757.

Liu, Haoxuan; Zhang, Jianzhi (2019): Yeast spontaneous mutation rate and spectrum vary with environment. In *Curr. Biol.* 29 (10), 1584-1591. e3.

Lynch, Michael; Ackerman, Matthew S.; Gout, Jean-Francois; Long, Hongan; Sung, Way; Thomas, W. Kelley; Foster, Patricia L. (2016): Genetic drift, selection and the evolution of the mutation rate. In *Nat. Rev. Genet.* 17 (11), pp. 704–714. DOI: 10.1038/nrg.2016.104.

Möller, Mareike; Habig, Michael; Lorrain, Cécile; Feurtey, Alice; Haueisen, Janine; Fagundes, Wagner C. et al. (2021): Recent loss of the Dim2 DNA methyltransferase decreases mutation rate in repeats and changes evolutionary trajectory in a fungal pathogen. In *PLoS Genet* 17 (3), e1009448. DOI: 10.1371/journal.pgen.1009448.

Möller, Mareike; Schotanus, Klaas; Soyer, Jessica L.; Haueisen, Janine; Happ, Kathrin; Stralucke, Maja et al. (2019): Destabilization of chromosome structure by histone H3 lysine 27 methylation. In *PLoS Genet* 15 (4), e1008093. DOI: 10.1371/journal.pgen.1008093.

Nguyen, Duong T.; Wu, Baojun; Long, Hongan; Zhang, Nan; Patterson, Caitlyn; Simpson, Stephen et al. (2020): Variable spontaneous mutation and loss of heterozygosity among heterozygous genomes in yeast. In *Mol. Biol. Evol.* 37 (11), pp. 3118–3130.

Schotanus, Klaas; Soyer, Jessica L.; Connolly, Lanelle R.; Grandaubert, Jonathan; Happel, Petra; Smith, Kristina M. et al. (2015): Histone modifications rather than the novel regional centromeres of *Zyoseptoria tritici* distinguish core and accessory chromosomes. In *Epigenetics & chromatin* 8, p. 41. DOI: 10.1186/s13072-015-0033-5.

Reviewers' Comments:

Reviewer #1:

Remarks to the Author:

Authors have adequately addressed my comments. I have no more comments.

Reviewer #2:

Remarks to the Author:

The authors did a nice job addressing the concerns of the reviewers and the manuscript has been significantly improved. It will be an important contribution to the field.

Reviewer #3:

None